# Causally-Enhanced Reinforcement Policy Optimization of Large Language Models

## Abstract

Large language models (LLMs) trained with reinforcement objectives often achieve superficially correct answers via shortcut strategies, pairing correct outputs with spurious or unfaithful reasoning and degrading under small causal perturbations. We introduce *Causally-Enhanced Policy Optimization* (CE-PO), a drop-in reward-shaping framework that augments policy optimization with a differentiable proxy for causal coherence along the generation pathway from prompt ($Z$) to rationale ($X$) to answer ($Y$). CE-PO estimates model-internal influence with Jacobian-based sensitivities, counterfactually hardens these signals to suppress nuisance cues, and fuses the resulting coherence score with task-accuracy feedback via a Minkowski (power-mean) combiner, exposing a single tunable between accuracy and coherence trade-off. The unified reward integrates with PPO/GRPO without architectural changes. Across reasoning benchmarks and causal stress tests, CE-PO reduces reward hacking and unfaithful chain-of-thought while improving robustness to correlation–causation flips and light counterfactual edits, all at near-parity accuracy. Experimental results across 4 datasets show that CE-PO improves accuracy over baselines by 5.49 % on average (up to 9.58 %), while improving robustness to correlation–causation flips and light counterfactual edits.

## 1 Introduction

Outcome-centric reinforcement training of large language models (LLMs)—whether via standard policy optimization or preference-based tuning (Christiano et al., 2017; Ouyang et al., 2022; Schulman et al., 2017; Shao et al., 2024) optimizes *what* answer is produced but places little pressure on *how* that answer is obtained (Sim et al., 2025). As a result, models routinely learn to "game" reward signals (e.g., by exploiting length, format, or lexical overlap) and to pair correct answers with unfaithful or spurious reasoning traces (Wen et al., 2024; Arjovsky et al., 2020). This failure mode is not merely theoretical: without intervention, we observe models can game both accuracy proxies and *gradient-based sensitivity* surrogates—e.g., by guessing final answer or maximizing output length as in section 3. This mismatch between scoring for outcomes and expecting coherent reasoning harms robustness, causing severe and threatening hallucination.

We argue that robust reasoning requires not only accurate outputs but also *causal coherence* along the generation pathway. By *coherence* we mean that (i) the prompt genuinely informs the rationale, (ii) the prompt along with rationale in turn genuinely informs the final answer, and (iii) small, causally relevant edits to the prompt or rationale induce correspondingly appropriate changes downstream, whereas irrelevant edits do not. Concretely, let ($Z \rightarrow X \rightarrow Y$) denote the intended chain from prompt ($Z$), to rationale ($X$), to answer ($Y$) (Bao et al., 2024a). If the model's internal influence respects this order, chain-of-thought (CoT) (Wei et al., 2022) traces are more likely to be right for the right reasons (Ouyang et al., 2022; Shao et al., 2024) and to generalize under distribution shift. However, current reinforcement objectives offer no internal signal that rewards such coherence, as previous literature on causal reward in LLM typically focus on causally indifferent to external factors instead of reasoning coherence (Wang et al., 2025). Related efforts to promote faithful reasoning typically rely on external supervision or decoding-time control: rationale supervision and "right-for-the-right-reasons" regularizers require human explanations and usually target only input and generation; concept-bottleneck (Koh and Liang, 2017) and invariance/IRM approaches (Kamath et al., 2021) enforce intermediate structure but do not address generative $prompt \rightarrow rationale \rightarrow answer$ flows; CoT verification and constrained decoding act *post hoc* (Chi et al., 2025; Heimersheim et al.,

2024) rather than shaping training. In contrast, we seek a *differentiable, annotation-free, training-time* signal that explicitly rewards influence coherence across $Z \to X \to Y$ and integrates into standard policy optimization.

Translating the above goal into a trainable signal raises three coupled issues. *Measurability*—existing faithfulness tools (saliency, raw input-gradient attributions) are fragile and can fail basic sanity checks or depend weakly on the learned model, limiting their usefulness as training signals (Adebayo et al., 2018); gradient regularization that makes models "right for the right reasons" typically relies on human-provided rationales and targets only input→output links in discriminative settings, not the generative prompt→rationale→answer pathway (Ross et al., 2017), while causal-reasoning benchmarks show LLMs still conflate correlation and causation (Jin et al., 2024). This motivates a *differentiable, model-internal* proxy for directed influence along $Z \to X$, $X \to Y$, and $Z \to Y$ that is compatible with policy-gradient optimization. *Robustness to spurious sensitivity*—naïve gradient signals are easily inflated by non-semantic factors (token frequency, position, format), encouraging shortcut learning (Geirhos et al., 2020) and *reward hacking* under outcome-only RL objectives (Amodei et al., 2016); moreover, CoT can appear compelling yet remain unfaithful and brittle under small perturbations (Wei et al., 2022; Bao et al., 2024b; Chi et al., 2024). Hence we require *counterfactual hardening* that explicitly breaks semantic links and filters out nuisance directions rather than trusting raw sensitivities. *Objective balance*—robustness-oriented methods (e.g., invariance penalties) often trade off in-distribution accuracy and can underperform ERM when mis-specified (Arjovsky et al., 2020; Rosenfeld et al., 2020; Kamath et al., 2021), whereas optimizing only task reward preserves shortcut exploitation (Amodei et al., 2016). We therefore need a *tunable* mechanism to balance accuracy against coherence during optimization, avoiding both over-regularization and reward hacking.

**Our approach.** We introduce C̲ausally-E̲nhanced Policy Optimzation(CE-PO), a drop-in reward-shaping scheme that adds a *Jacobian-based causal-coherence signal* to standard policy optimization and fuses it with task accuracy via a *Minkowski (power-mean) combiner*. Concretely, we compute model-internal Jacobian influence signals for the $prompt \to rationale$ and $rationale \to answer$ links; harden them with a simple counterfactual procedure (removing the resulting nuisance directions by reshuffling) to reduce sensitivity to superficial cues; normalize and aggregate into a single *coherence score*; and combine this score with accuracy using a power-mean with tunable weights/exponent, yielding a single reward that trades off coherence and accuracy. The unified reward is plugged into PPO/GRPO without architectural changes. In summary, our contributions are threefold:

- We propose a differentiable, model-internal *DCE-proxy* (influence–coherence) reward along $Z \to X \to Y$, coupled with *counterfactual residualization* (see subsection 2.1) to suppress shortcut sensitivities and *reward hacking*.
- We introduce a *Minkowski (power-mean) combiner* (detailed in subsection 2.2) that exposes a tunable trade-off between task accuracy and causal-coherence signals, serving as a drop-in objective for PPO/GRPO with efficient reward computation.
- CE-PO demonstrates consistent performance improvements, supported by comprehensive ablation analyses and competitive out-of-distribution generalization. In addition, our study highlights notable generation patterns and offers insights into model behavior.

## 2 CAUSAL ENHANCED POLICY OPTIMIZATION

**Overview.** In this section, we give implementation details of *CE-PO* and formalize it. Given a rollout with prompt $Z$, rationale $X$, and answer $Y$, as shown in Figure 1. The method unfolds in three steps that map one-to-one to the subsections below. (i) **Raw influence signals** : we compute *Base Jacobians*, i.e., local sensitivities of downstream span likelihoods with respect to upstream token embeddings, as differentiable, model-internal proxies for $Z \to X$, $X \to Y$, and for $Z \to Y$ removing mediator $X$. (ii) **Counterfactual hardening** : Deploying raw Jacobian scores as rewards risks *reward hacking*, where spurious factors (e.g., length) dominate the signal Figure 2. To suppress such shortcut and formatting sensitivities, we break semantic links (reshuffle/mismatch) and remove the induced nuisance directions from the raw signals. (iii) **Reward normalization and fusion** We normalize Jacobian responses into a stable scalar *coherence score* and then fuse it with task accuracy using a Minkowski (power-mean) combiner, yielding a unified reward that exposes a tunable accuracy–coherence trade-off and is optimized with PPO/GRPO. This section defines each component, provides complexity notes, and clarifies how they compose into the training loop.

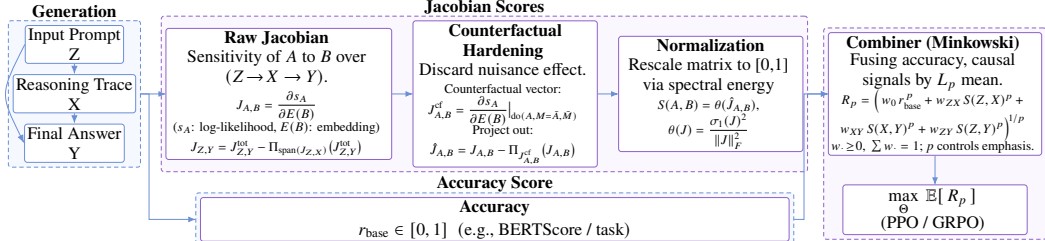

Figure 1: **CE-PO pipeline (left → right).** *Generation:* $Z \to X \to Y$ (X/Y split by a special token; see Fig. 14). *Base Jacobians:* local sensitivities for $Z{\to}X$, $X{\to}Y$, $Z{\to}Y$. *Counterfactual hardening:* build source-side counterfactuals and project their subspace from the base signals to obtain hardened residuals. *Normalization:* use spectral energy to convert Jacobian matrix into [0,1] scalar. *Fusion:* combine $S(A, B)$ signals with accuracy $r_{\text{base}}$ via a Minkowski combiner; optimize the unified reward with PPO/GRPO.

## 2.1 JACOBIAN SIGNALS, AND CAUSALLY-ENHANCED REWARD

We consider a differentiable language model $\Theta$ and our goal is, as previously described, to improve the accuracy on $Y$ (i.e., $r_{base}$) yet simultaneously to ensure the alignment of influence with this order: $Z$ should shape $X$, and $X, Z$ should shape $Y$. Our proposed method is illustrated in Figure 1, providing a causally grounded reward signal $R_p$ by combining $r_{base}$ and Jacobian-based scores measuring causal effects along $Z \to X \to Y$.

**Base Jacobians.** While $Y$-accuracy rewards *what* answer is produced, it does not constrain *how* the answer is obtained. To shape the reasoning pathway, we introduce a differentiable, model-internal *Direct Causal Effect (DCE) proxy* (Pearl, 2009; VanderWeele, 2011) for coherence along $Z \to X \to Y$: small and causal changes in upstream tokens should induce appropriate changes downstream. Concretely, let $s_X$ and $s_Y$ denote the log-likelihoods of the rationale span $X$ and the answer span $Y$ posterior to $Z$ and $Z, X$ as in Equation 1, and let $E(A) \in \mathbb{R}^{T_A \times d}$ be the token-embedding matrix for a text segment $A$ (length $T_A$, hidden size $d$) and $a_{<t}$ denotes the prefix strictly before index $t$.

$$s_X = \sum_{t \in X} \log p_\Theta(x_t \mid Z, x_{<t}), \qquad s_Y = \sum_{t \in Y} \log p_\Theta(y_t \mid Z, X, y_{<t}) \tag{1}$$

We measure the influence from $A$ to the score of $B$ by the *block Jacobian* $J_{AB} \triangleq \partial s_B / \partial E(A)$. For the intended chain $Z \to X \to Y$, the *Base Jacobians* (raw, unhardened sensitivities) are

$$J_{ZX} = \frac{\partial s_X}{\partial E(Z)} \in \mathbb{R}^{T_Z \times d}, \qquad J_{XY} = \frac{\partial s_Y}{\partial E(X)} \in \mathbb{R}^{T_X \times d}, \qquad J_{ZY}^{\text{total}} = \frac{\partial s_Y}{\partial E(Z)} \in \mathbb{R}^{T_Z \times d}. \tag{2}$$

In addition, to approximate the Direct Causal Effect (DCE) from $Z$ to $Y$, we need to isolate X effect out of $J_{ZY}^{\text{total}}$. We remove the mediated path via projection: forming a via-$X$ template from $J_{ZX}$ ($Z \to X$) and the principal direction of $J_{XY}$ ($X \to Y$), then subtracting its projection from the total effect to obtain the direct effect:

$$J_{ZY} = J_{ZY}^{\text{total}} - \Pi_{J_{ZX} \odot \bar{u}}\left(J_{ZY}^{\text{total}}\right), \tag{3}$$

where $\bar{u} = \frac{1}{T_X} \sum_{t \in X} \frac{J_{XY}[t,:]}{\|J_{XY}[t,:]\|_2}$, and $\Pi_A(B) = \frac{\langle B, A \rangle_F}{\langle A, A \rangle_F} A$, with $\langle B, A \rangle_F$ indicating the Frobenius inner product between $A$ and $B$, and $\odot$ be Hadamard product.

We use the Jacobian as a local, direct causal effect *proxy*: for any upstream–downstream pair $(A, B) \in \{(Z, X), (X, Y), (Z, Y)\}$, with mediators held at their observed values, $J_{AB} = \partial s_B / \partial E(A)$ quantifies how an infinitesimal perturbation to $A$'s embeddings shifts the downstream log-likelihood $s_B$—i.e., which directions in $A$ immediately move $B$, and by how much. Crucially, $J_{AB}$ is not equal to a causal DCE in the mediation sense; it is the *first-order Taylor approximation* around the observed point (Pearl, 2001). They are already informative enough and avoid the cost and instability of higher-order (Hessian/influence-function) estimators in deep/LLM settings (Pearlmutter, 1994; Martens, 2010; Koh and Liang, 2017; Li et al., 2024).

We use $J_{AB}$ to serve as the starting point for our *coherence* signal. In the next subsections we apply *counterfactual hardening* to remove nuisance directions and then normalize/aggregate the residual influence into a scalar coherence score that is fused with task accuracy.

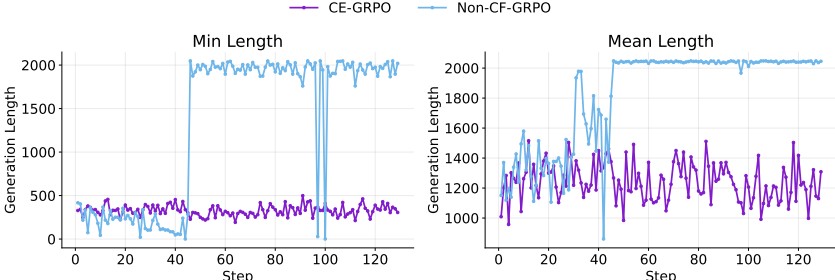

Figure 2: Training trajectories comparison: **Non-CF-GRPO** (GRPO with non-CF award) vs **CE-GRPO** (GRPO with our CE award) on Qwen-3-4B-Thinking with `max_tokens`=2048, in terms of minimum (left) and mean (right) generation length. Non-CF-GRPO curves plateau as generations approach the token limit, indicating length-driven reward hacking, while CE-GRPO doesn't.

**Counterfactual-hardened Jacobian score.** Jacobian-based rewards, when computed directly on the base input, may conflate causal content with nuisance factors such as style, formatting and token-frequency statistics. As a result, innocuous paraphrases or vacuous elongation can inflate the gradient norm and concentrate its top singular direction, enabling *length-driven reward hacking* up to the token limit (Gao et al., 2023; Rafailov et al., 2024). Empirically, as shown in Figure 2 (blue), the length hacking existing in non-counterfactual Jacobian rewards (Non-CF-GRPO conflates).

To suppress shortcut-driven sensitivities in the *Base Jacobians*, we harden them with pair-specific counterfactuals. For each link $(A, B) \in \mathcal{L} := \{(Z, X), (X, Y), (Z, Y)\}$, let $M(A, B)$ denote mediators on $A{\to}B$ paths (empty except $M(Z, Y) = \{X\}$). We break the semantic alignment between $A$ (and $M$) and $B$ by permuting the tokens of $A$ (and of $M$ when present), yielding $(\bar{A}, \bar{M})$ that preserves surface statistics (length/frequency) while being approximately independent of $B$. Let $s_B$ be the log-likelihood of span $B$ and $E(A) \in \mathbb{R}^{T_A \times d}$ the embedding matrix of $A$. Token reshuffling is used to construct nuisance dimensions because we target to destroy semantics while perfectly preserving low level nuisance cues, comparing to alternative methods like deleting or replacing, etc. We compute the *counterfactual Jacobian*

$$J_{AB}^{\text{cf}} = \left.\frac{\partial s_B}{\partial E(A)}\right|_{(A,M)\mapsto(\bar{A},\bar{M})}. \tag{4}$$

We construct $K$ *source-side* counterfactuals and calculate their Jacobians by Equation 4, which are then averaged to have $\bar{J}_{AB}^{\text{cf}}$ (we set $K$=4, larger values generally improve results but incur higher computational cost).

Then, we extract the principal counterfactual subspace via the top-$k$ left singular vectors $U_{AB}$ of $J_{AB}^{\text{cf}}$, define the projector $\Pi_{AB}(J) = U_{AB} U_{AB}^\top J$, following (Ravfogel et al., 2020). Taking the $J_{AB}$ in Equation 2 and Equation 3, we obtain the deconfounded (direct) block by removing these directions:

$$\widehat{J}_{AB} = J_{AB} - \Pi_{AB}(J_{AB}). \tag{5}$$

Permuted signals expose a spurious nuisance subspace; projecting it out yields a deconfounded, causally aligned signal that resists surface correlations and reward hacking (Ravfogel et al., 2020).

As shown in Figure 2 (purple), the Non-CF-GRPO baseline drives the mean generation length to the token cap, revealing a length-based reward-hacking loop. With our counterfactual reshuffling and projection (CE-GRPO), the curve stays well below the cap and does not plateau, while reward improves. The mechanism is simple: source-side counterfactuals preserve surface statistics (length-/frequency) while scrambling semantics (Ravfogel et al., 2020), so the counterfactual Jacobian spans a nuisance subspace; orthogonally projecting the base Jacobian off this subspace—akin to Neyman orthogonalization (Oprescu et al., 2019)—decouples the reward from length/format sensitivities and correlations (Chernozhukov et al., 2018). For the $Z{\to}Y$ link, we additionally subtract the via-$X$ component before projection (a first-order direct-effect correction), further reducing spurious incentives.

**Reward normalization and fusion** Finally, we *scalarize* the block to a score via the scale-free spectral energy share $S(A, B) = \phi(\widehat{J}_{AB})$, where $\phi(J) = \sigma_1(J)^2/\|J\|_F^2 \in [0, 1]$ to squeeze and standardize the process (Vershynin, 2018). Concretely,

$$S(Z, X) = \phi(\widehat{J}_{ZX}), \qquad S(X, Y) = \phi(\widehat{J}_{XY}), \qquad S(Z, Y) = \phi(\widehat{J}_{ZY}). \tag{6}$$

Let $r_{\text{base}} \in [0, 1]$ be a task accuracy score on $Y$ (e.g., semantic correctness, measured by BERTScore (Zhang et al., 2019) in our setting). We define the new causally grounded reward by integrating accuracy and the three counterfactual-enhanced signals via a weighted power mean (Minkowski combiner) (Bullen, 2013),

$$R_p = \left( w_0 \, r_{\text{base}}^{p} + w_1 \, S(Z, X)^p + w_2 \, S(X, Y)^p + w_3 \, S(Z, Y)^p \right)^{1/p}, \tag{7}$$

with $p \in \mathbb{R}$, weights $w_i \geq 0$, and $\sum_{i=0}^{3} w_i = 1$. As $p \to 0$, $R_p$ approaches a weighted geometric mean (minimal optimization); larger $p$ emphasizes the largest channel, which respectively matches deconfounded Jacobian causal scores and BERTScore as in subsection 2.2. Using this $R_p$ in any common policy-gradient training of LLM (e.g., PPO, GRPO), the goal is to maximize $\mathbb{E}[R_p]$.

*Note of $r_{base}$:* We set $r_{\text{base}}$ by BERTScore (i.e., the cosine similarity between $Y$ and ground truth in BERT embedding space), as it is a continuous measure between 0 and 1 for accuracy, aligning with the scale of the causal term. In contrast, 0/1 judgments are often sensitive to judging prompts and are less compatible; mixing such binary signals with the causal term can induce oscillatory updates. Ablation studies on replacing BERTScore with 0/1 rewards are provided in section 3. In addition, we also provide TRPO-form (Schulman et al., 2015) lower bound for CE-PO with details in Appendix B and the discussion for training efficiency can be found in Appendix C.

**Training objective** Let $R_p(Z, X, Y) \in [0, 1]$ denote the unified reward from Eq. (7). We treat $R_p$ as a *scalar terminal reward* for each rollout $(Z, X, Y)$: $r_T = R_p$, $r_t = 0$ for $t < T$, and **do not backpropagate through** $R_p$ (standard policy-gradient; the Jacobian signals are used only to compute $R_p$).

**CE-PPO.** With behavior policy $\pi_{\text{old}}$, importance ratio $w_t(\theta) = \pi_\theta(a_t|s_t)/\pi_{\text{old}}(a_t|s_t)$, and GAE advantages $\hat{A}_t$ computed from the sparse reward, the clipped objective is

$$\mathcal{J}_{\text{CE-PPO}}(\theta) = \mathbb{E}\Big[ \min\big(w_t(\theta)\hat{A}_t, \, \text{clip}(w_t(\theta), 1-\epsilon, 1+\epsilon)\hat{A}_t\big) - \beta \, \text{KL}\big(\pi_\theta \,\|\, \pi_{\text{ref}}\big) \Big]. \tag{8}$$

We optionally standardize $R_p$ per batch to stabilize advantages.

**CE-GRPO.** For each prompt we sample $K$ candidates, compute $R_{p,k}$, and form group-relative advantages $\tilde{A}_k = \big(R_{p,k} - \mu_{\text{grp}}\big)/(\sigma_{\text{grp}} + \varepsilon)$, with $\mu_{\text{grp}}, \sigma_{\text{grp}}$ the group mean/std. The GRPO-style objective is

$$\mathcal{J}_{\text{CE-GRPO}}(\theta) = \mathbb{E}\Big[ \frac{1}{K} \sum_{k=1}^{K} \min\big(w_k(\theta)\tilde{A}_k, \, \text{clip}(w_k(\theta), 1-\epsilon, 1+\epsilon)\tilde{A}_k\big) - \beta \, \text{KL}\big(\pi_\theta \,\|\, \pi_{\text{ref}}\big) \Big]. \tag{9}$$

## 2.2 REWARD SIGNALS TRADE-OFF

Our RL objective (maximizing $\mathbb{E}[R_p]$) blends two signals, semantic accuracy (e.g., BERTScore) and causal coherence (counterfactual-enhanced Jacobian). These signals misalign at times, mirroring RLHF's trade-off between surface correctness and shortcut avoidance (Kalai et al., 2025; Lightman et al., 2023). By defining $R_p$ via the Minkowski norm, we enable a dynamically tunable balance between the two signals.

Figure 3 illustrates how the Minkowski $p$-norm interpolation provides a smooth mechanism to mediate the trade-off between semantic accuracy and causal coherence. As $p \to +\infty$, the reward collapses to the larger signal, typically BERTScore, which is unprojected and numerically higher. This drives accuracy-focused optimization (as observed in all models). Conversely, as $p \to -\infty$, the reward collapses to the smaller signal (e.g., when $p = 0$ the interpolation

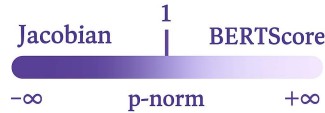

Figure 3: Tunable balance enabled by Minkowski $p$-norm interpolation in $R_p$.

equals geometric mean), which is usually the counterfactual Jacobian, lowered by residualization. This, in turn, emphasizes causal coherence. This multi-objective behavior parallels prior work on balancing conflicting RL rewards (Won et al., 2019; Roijers et al., 2013). To illustrate how CE-PO combine both signals can thus enhance LLM reasoning capability to be both causally capable yet not template-following, we provide an LLM generation example in Figure D.

## 3  EXPERIMENTS

In this section, we show performance of CE-PO on top of PPO and GRPO (noted as **CE-PPO** or **CE-GRPO**) against simple baselines, using BERTScore as the reward (denoted as PPO-BERTScore or GRPO-BERTScore) and vanilla LLMs, across representative LLMs and standard evaluation datasets.

**Experiment Setting.**   Training is performed on a single node with four A100 or H100 GPUs. Unless stated otherwise, we use a rollout group size of 6 per prompt, KL penalty of $\lambda_{\mathrm{KL}} = 0.001$ to stabilize optimization, and cap decoder outputs at $2048$ tokens and input within $512$ tokens. In determining $R_p$, extensive experiments indicate that a good configuration could be $p = 0.2$ and BERTScore, Jacobian scores weights, respectively, as $(w_0, w_1, w_2, w_3) = (0.7, 0.1, 0.1, 0.1)$, which is set as default for comparison unless otherwise specified in ablation studies. We set learning rate of $2 \times 10^{-6}$, batch size 32, and training for 10 epochs with temperature 0.6. Validation interval is set as 10 and checkpoints are saved per 100 steps. Experiments are based on Verl (Sheng et al., 2024) framework with mild modification for separate thread for reward calculation, and we also set warm-up step = 10 for critic.

To avoid random guesses and generate $X$ and $Y$, all models are prompted to generate CoT reasoning (Wei et al., 2022). Evaluation is reported as accuracy, and the response correctness is determined via an LLM-as-Judge protocol (Zheng et al., 2023) using GPT-4o-mini (see prompt in Figure 15, which is designed to distinguish guess or contradictions especially for multi-choice questions).

We evaluate both standard instruction-tuned LLMs and "thinking" variants that emit explicit intermediate reasoning and instruction following. Although compute constraints preclude experiments on 70B-scale models, our evaluation covers compact to mid-sized systems that are widely used in research: Llama-3-8B-Instruct, Llama-3.2-3B-Instruct, Phi-3.5-mini-Instruct, Qwen3-4B-Thinking, and Qwen3-1.7B-Thinking (Meta AI, 2024b;a; Microsoft, 2025; Qwen Team, 2025b;a). [1]

**Training and Validation Dataset.**   We train and tune on four reasoning-heavy sets: *BBEHCausal*, the causal reasoning split from BIG-Bench Extra Hard dataset, including counterfactual deduction or cause inference questions; (Google DeepMind, 2025)); *CaseHOLD*, tasks centering on multiple-choice legal holdings given case citation context (CaseHOLD, 2022); *MATHHARD*, the level-5 subset targeting multi-step mathematical reasoning with multiple-choice questions (Hendrycks et al., 2021); and *IfQA*, an open-domain QA under counterfactual *if*-clause presuppositions requiring hypothetical reasoning (Yu et al., 2023). Note that all datasets are formatted as question answer pairs, examples of them are presented respectively in Table 6 and Table 7.

**Testing Dataset.**   For same-field evaluation, we use *BBEHMATH* (challenging multistep arithmetic), where we filter shorter and more trivial questions and normalize alphabetic numerals to digits with unfolded calculation for comparability on this dataset (Google DeepMind, 2025). We further assess generalization with *CLadder*, covering association/intervention/counterfactual queries (Causal-NLP, 2023), *LegalBench* (case-understanding split) (Guha et al., 2023) analyzing the cause or reason of legal case, and *LogiQA*, a multiple-choice dataset emphasizing deductive logical reasoning sourcing from officer entrance exams (lucasmccabe, 2021).

**Main Experiment Results.**   Under the training and validation dataset split detailed above, Figure 4 shows CE-PPO delivering smooth, near-monotonic validation gains across five backbones and four validation sets: rapid early improvement followed by mild saturation, which is indicative of stable and standard RL optimization plot rather than overfitting. In Table 1, across all five backbones, both causal variants outperform their non-causal baselines: CE-PPO/CE-GRPO improve the best non-CE baseline by 2.3 to 9.58 points. Comparing CE-GRPO with the vanilla baseline, we observe gains of 5.8–6.0 points on Llama-3-8B and Llama-3.2-3B; the largest improvement appears on Qwen-3-1.7B-Thinking ($\approx 10$ points), whereas Phi-3.5-mini-Instruct shows the most marginal uplift. These patterns suggest that architectural and scale differences modulate the benefits of CE-GRPO. CE-GRPO is *not* uniformly stronger than CE-PPO, while slightly higher on Qwen-1.7B/4B, Phi-3.5-mini, and Llama-3-8B, it lags behind CE-PPO on Llama-3.2-3B, indicating complementary strengths rather than strict dominance.

---

[1]Note that all experiments use Instruct or Thinking models, as our outputs must be cleanly formatted into $X$ and $Y$, which base models struggle to separate. Since Instruct models are typically harder to improve with RL, the observed gains further highlight the effectiveness of CE-PO.

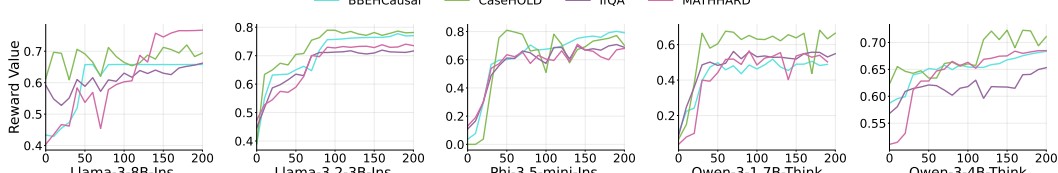

Figure 4: Reward curves of CE-PPO training on validation datasets across five models.

Table 1: **Main results across models, RL methods, and datasets.** We report LLM-as-Judge accuracy (%) under different RL variants: vanilla model , standard PPO/GRPO with BERTScore reward, and causal-enhanced (CE-PPO/CE-GRPO). We repeatedly generate 4 times of each trained model, and thus report mean and deviation below.

| Model | RL Method | Dataset (%) | | | | Average |
| --- | --- | --- | --- | --- | --- | --- |
| | | **BBEHMATH** | **CLadder** | **LegalBench** | **LogiQA** | |
| **Qwen-3-1.7B-Thinking** | Vanilla | $1.36 \pm 1.96$ | $34.77 \pm 0.68$ | $57.80 \pm 2.59$ | $50.00 \pm 0.45$ | 35.98 |
| | PPO-BERTScore | $5.91 \pm 0.00$ | $37.27 \pm 0.91$ | $58.26 \pm 1.95$ | $52.50 \pm 0.00$ | 38.48 |
| | GRPO-BERTScore | $3.64 \pm 1.29$ | $35.00 \pm 0.52$ | $60.67 \pm 0.94$ | $57.04 \pm 0.68$ | 39.09 |
| | CE-PPO | $\mathbf{10.45 \pm 0.52}$ | $\mathbf{49.32 \pm 0.68}$ | $60.67 \pm 1.05$ | $\mathbf{58.05 \pm 1.14}$ | 44.62 |
| | CE-GRPO | $10.00 \pm 1.29$ | $44.77 \pm 0.00$ | $\mathbf{70.64 \pm 0.00}$ | $56.82 \pm 0.45$ | **45.56** |
| **Qwen-3-4B-Thinking** | Vanilla | $6.82 \pm 0.45$ | $42.95 \pm 0.91$ | $74.77 \pm 1.95$ | $80.45 \pm 0.68$ | 51.25 |
| | PPO-BERTScore | $5.00 \pm 1.05$ | $45.00 \pm 0.00$ | $73.64 \pm 0.47$ | $80.00 \pm 0.45$ | 50.91 |
| | GRPO-BERTScore | $6.82 \pm 0.45$ | $40.68 \pm 0.68$ | $76.61 \pm 0.65$ | $83.86 \pm 0.91$ | 51.99 |
| | CE-PPO | $\mathbf{9.55 \pm 0.52}$ | $45.45 \pm 0.68$ | $\mathbf{79.34 \pm 1.05}$ | $79.77 \pm 0.91$ | 53.53 |
| | CE-GRPO | $7.27 \pm 0.00$ | $\mathbf{49.09 \pm 0.64}$ | $76.90 \pm 0.45$ | $\mathbf{84.09 \pm 0.45}$ | **54.34** |
| **Phi-3.5-mini-Ins.** | Vanilla | $6.13 \pm 0.52$ | $40.23 \pm 0.68$ | $68.35 \pm 1.95$ | $54.09 \pm 0.45$ | 42.20 |
| | PPO-BERTScore | $6.59 \pm 0.00$ | $43.40 \pm 0.87$ | $66.97 \pm 0.00$ | $54.77 \pm 0.00$ | 42.93 |
| | GRPO-BERTScore | $5.45 \pm 0.87$ | $41.59 \pm 0.64$ | $68.81 \pm 1.30$ | $52.73 \pm 0.45$ | 42.15 |
| | CE-PPO | $\mathbf{8.41 \pm 0.00}$ | $42.95 \pm 0.91$ | $70.18 \pm 1.95$ | $55.90 \pm 0.68$ | 44.36 |
| | CE-GRPO | $6.81 \pm 0.64$ | $\mathbf{45.45 \pm 0.45}$ | $70.18 \pm 0.65$ | $\mathbf{57.95 \pm 0.00}$ | **45.10** |
| **Llama-3.2-3B-Ins.** | Vanilla | $3.63 \pm 0.91$ | $30.00 \pm 0.68$ | $45.41 \pm 0.47$ | $41.59 \pm 0.45$ | 30.16 |
| | PPO-BERTScore | $6.59 \pm 0.68$ | $33.63 \pm 0.64$ | $45.87 \pm 0.00$ | $38.41 \pm 0.45$ | 31.12 |
| | GRPO-BERTScore | $6.50 \pm 0.45$ | $31.36 \pm 0.91$ | $42.73 \pm 2.57$ | $44.09 \pm 0.00$ | 31.17 |
| | CE-PPO | $\mathbf{10.67 \pm 1.14}$ | $\mathbf{43.64 \pm 0.68}$ | $\mathbf{55.05 \pm 1.30}$ | $\mathbf{48.18 \pm 0.91}$ | **39.38** |
| | CE-GRPO | $7.95 \pm 0.91$ | $35.23 \pm 0.00$ | $54.13 \pm 2.59$ | $47.27 \pm 0.45$ | 36.15 |
| **Llama-3-8B-Ins.** | Vanilla | $7.05 \pm 0.45$ | $39.54 \pm 0.91$ | $61.93 \pm 1.95$ | $65.68 \pm 0.64$ | 43.55 |
| | PPO-BERTScore | $5.27 \pm 0.64$ | $37.95 \pm 0.00$ | $61.93 \pm 0.00$ | $60.91 \pm 0.45$ | 41.52 |
| | GRPO-BERTScore | $10.91 \pm 1.14$ | $39.09 \pm 0.91$ | $57.80 \pm 0.00$ | $67.50 \pm 0.00$ | 43.83 |
| | CE-PPO | $\mathbf{14.86 \pm 1.59}$ | $44.77 \pm 0.45$ | $\mathbf{67.35 \pm 0.79}$ | $68.40 \pm 0.91$ | 48.84 |
| | CE-GRPO | $13.64 \pm 0.91$ | $\mathbf{45.00 \pm 0.64}$ | $66.97 \pm 0.00$ | $\mathbf{71.82 \pm 0.45}$ | **49.36** |

**Out of Distribution(OOD) Evaluation.** Trained exclusively on causal- and math-reasoning corpora, our CE-PO models are stress-tested for OOD generalization on TruthfulQA (Lin et al., 2021) (misconception resistant factuality), CodeMMLU (Manh et al., 2024) (programming knowledge), and SuperGPQA (Du et al., 2025) (expert level STEM QA), evaluating 200 uniformly random samples from each benchmark. As shown in Figure 5, they deliver consistent—but task-dependent improvements over baselines: effects are strongest on structure-heavy code reasoning, moderate on STEM-oriented SuperGPQA, and smaller yet steady on TruthfulQA, suggesting that causal regularization chiefly benefits multi-step reasoning while preserving overall robustness.

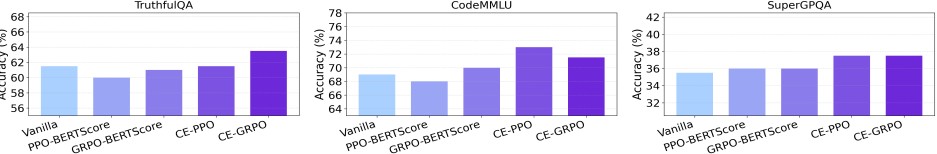

Figure 5: Qwen-3-4B-Thinking accuracy in OOD evaluation scenarios.

**Ablation Studies.** We test CE-PO's composite objective under a fixed setup (Llama-3-8B-Instruct, GRPO, max_tokens=2048; datasets: BBEHMath, CLadder, LegalBench, LogiQA). The base reward is $r_{\text{base}}$ (BERTScore); coherence uses hardened link scores $S(Z, X), S(X, Y), S(Z, Y)$. We ablate

Table 2: Ablation Studies on Llama-3-8B-Ins.

| Ablation | BBEHMATH | CLadder | LegalBench | LogiQA | Average |
|---|---|---|---|---|---|
| $p = 10$ | 6.36 | 39.09 | 56.90 | 67.27 | 42.91 |
| $p = -2$ | 12.73 | 43.63 | 65.52 | 70.00 | 47.97 |
| $p = 1$ | 10.00 | 50.00 | 60.34 | 64.55 | 46.22 |
| $w=[0, 1/3, 1/3, 1/3]$ | 10.91 | 44.54 | 61.93 | 68.18 | 46.89 |
| $w=[4/5, 1/10, 1/10, 0]$ | 5.45 | 23.63 | 40.37 | 41.82 | 27.82 |
| $w=[4/5, 1/10, 0, 1/10]$ | 3.63 | 30.00 | 45.87 | 49.09 | 32.15 |
| $w=[4/5, 0, 1/10, 1/10]$ | 5.45 | 31.82 | 43.12 | 40.90 | 30.32 |
| $w=[1/2, 1/6, 1/6, 1/6]$ | 8.18 | 41.82 | 67.24 | 66.36 | 45.90 |
| Perturbed Reward | 7.27 | 40.00 | 65.52 | 69.09 | 45.97 |
| Judge Reward (binary) | 6.36 | 39.09 | 60.34 | 66.36 | 43.04 |
| **CE-GRPO** | 13.64 | 47.27 | 66.97 | 72.73 | 50.65 |

the Minkowski exponent $p$ and weights $w$, drop individual links, add Gaussian noise to $r_{\text{base}}$, and swap BERTScore for an LLM-as-Judge. Ablation results in Table 2 yield:

**Insight 1:** Reward hacking would emerge once we remove any of $S(Z, X)$, $S(Z, Y)$ and $S(X, Y)$. Models under such flawed RL can't even excel the original models as observed in row 5 to 7 in Table 2. The likely cause is an incomplete constraint, i.e., the causal loop $Z \rightarrow X \rightarrow Y$ is not closed, so the optimization ceases to be strictly benign. With a detailed supporting plot provided in Figure 11 of the reward signal and generation length, we can observe that initial removal breaks the prior balance (reward dips), and subsequent updates discover a length shortcut (length conflating).

**Insight 2:** By selecting different $p$ values, we can observe the performances of $p$ as 10 identical to BERTScore only and $p$ as -2 to Jacobian scores only (i.e. $w$ as $[0,1/3, 1/3, 1/3]$). The same case of $p$ as 1, which reveals that tuning $p$ can dynamically tune the optimization objective.

**Insight 3:** Under a perturbed reward (Gaussian noise $\sigma=0.05$), the model maintains satisfactory performance, indicating robustness of the training design. Using GPT - 4o - mini as an LLM-as-Judge (vs. BERTScore) yields performance similar to the original model, revealing what's prescribed in the *Note of $r_{base}$* in subsection 2.1.

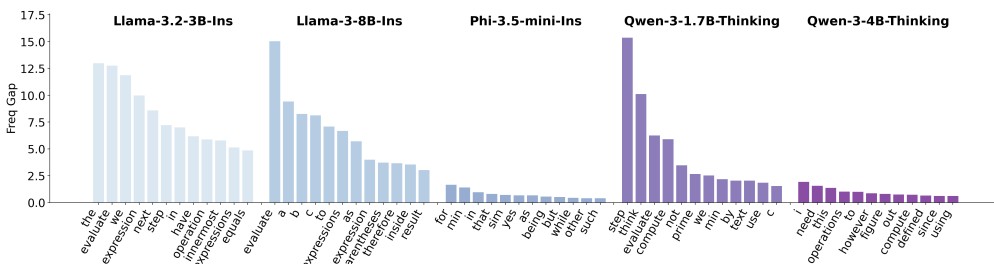

Figure 6: Frequency gap of generation per word where CE-GRPO exceeds BERTScore and Vanilla LLM (Top-12 per model).

**Generation Pattern Analysis.** Figure 6 shows that CE-GRPO does not induce uniform gains across all tokens but selectively amplifies reasoning or causal related words, suggesting that causal regularization guides the model toward tokens central to stepwise inference rather than surface templates. Complementarily, Figure 7 reveals the structural relation between metrics: $S(Z, X)$ is nearly independent of $S(X, Y)$ and $S(Z, Y)$, while the latter two are moderately correlated. This indicates that "evidence infusion" and "answer stability" provide orthogonal yet complementary signals. Their joint use, as in CE-GRPO, therefore supports coherence across the full $Z \rightarrow X \rightarrow Y$ pathway, rather than relying on any single proxy.

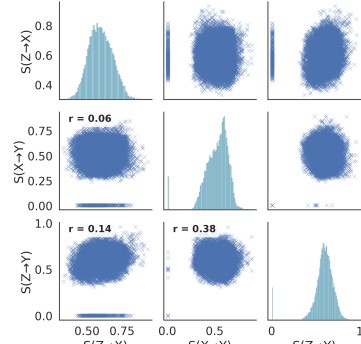

Figure 7: Pairwise scatter matrix of causal metrics on Qwen-3-4B-Thinking.

Turning to sequence-level behavior, Figure 9 reports the response lengths of CE-GRPO, GRPO-BERTScore, and the vanilla model on the training sets. The

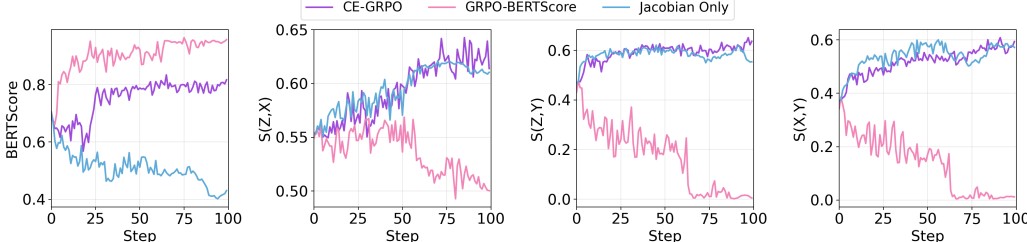

Figure 8: Training trajectories on Llama-3-8B-Instruct, reporting BERTScore, S(Z, X), S(Z, Y), S(X, Y)(left to right). Comparison is made between CE-GRPO and optimization with only causal signal (Jacobian) or accuracy signal (BERTScore), supporting the claim of reward signal balance in subsection 2.2.

means are closely matched, indicating our optimization is largely *length-indifferent*. CE-GRPO achieves better task performance without increasing the token budget beyond the baseline, suggesting the efficiency of CE-GRPO.

Finally, the training reward trajectories in Figure 8 highlight three regimes: BERTScore-only optimization inflates BERTScore only while collapsing causal channels, Jacobian-only boosts sensitivities but remains unstable and harms task accuracy. This finding echoes prior work showing that proxy metric optimization (e.g., ROUGE) misaligns with human preferences (Stiennon et al., 2020), while self-rewarded training inflates output length, underscoring the tension between verifiable rewards and reward-gaming behaviors (Yuan et al., 2025). CE-GRPO achieves smooth improvements across $S(Z, X)$, $S(X, Y)$, and $S(Z, Y)$ while keeping BERTScore competitive. This

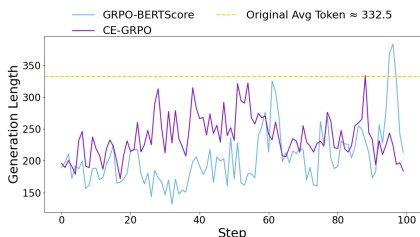

Figure 9: Response length comparison on Llama-3-8B-Ins

pattern underscores CE-GRPO's ability to sustain the causal chain from prompt to reasoning to answer, offering a practical safeguard against shortcut learning.

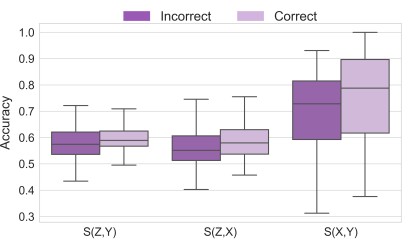

Figure 10: Boxplots of $S(Z, X)$, $S(Z, Y)$, and $S(X, Y)$ for LLM-as-Judge accuracy regarding their causal trajectory.

**Jacobian Signals and Accuracy** Figure 10 shows the distributions of Jacobian-based scores for samples annotated by correctness with LLM-as-Judge with strict judging criteria provided in Figure 15 conducted by GPT-4o. Annotations were collected on 1000 samples generated by Llama-3-8B-Ins. In total, 371 cases were labeled as consistent and 629 as inconsistent. To assess differences between the two groups, we conducted t-test with results reported in Table 5. We evaluate whether higher $S(A, B)$ scores align with factual correctness, as validity of these causal signals requires separation under strict accuracy criteria. As in Table 5, all three metrics show significantly higher means for the correct group ($p < 0.01$), confirming that $S(A, B)$ faithfully tracks reasoning–answer validity beyond surface outputs, and by statistically aligns with cognitive accuracy.

**Experiments on CE-PO's Scalability for** $r_{base}$ Generally, the Minkowski combiner (Equation 7) benefits from a continuous $r_{base} \in [0, 1]$ that fuses smoothly with our dense causal signals (Appendix C). BERTScore offers such a graded, semantics-aware signal and avoids the oscillatory behavior of discrete 0/1 rewards (e.g., LLM-as-Judge in Table 2). To show that CE-PO fusion is not limited to BERTScore-style similarity rewards, we further replace $r_{base}$ with a DeBERTa-v3-large reward model, representing classifier-style accuracy feedback. As summarized in Table 3, CE-PO maintains strong performance across both reward types using the same backbones and datasets as Table 1. Replacing BERTScore with pretrained reward model also doesn't introduce very significant difference, suggesting these two signals are both reliable and feasible.

| Model | Method | BBEHMATH | CLadder | LegalBench | LogiQA | Average |
|-------|--------|----------|---------|-----------|--------|---------|
| Qwen-3-4B-Thinking | GRPO-RM | 6.61 | 44.44 | 77.33 | 81.03 | 00.00 |
| | CE(RM)-GRPO | 8.39 | 52.37 | 77.15 | 83.70 | 00.00 |
| Phi-3.5-mini-Ins. | GRPO-RM | 00.00 | 00.00 | 00.00 | 00.00 | 00.00 |
| | CE(RM)-GRPO | 00.00 | 00.00 | 00.00 | 00.00 | 00.00 |
| Llama-3-8B-Ins. | GRPO-RM | 8.39 | 39.54 | 63.00 | 69.58 | 00.00 |
| | CE(RM)-GRPO | 14.79 | 45.00 | 67.35 | 72.62 | 00.00 |

Table 3: Benchmarking comparison across RM (pretrained DeBERTa reward model) RL method and CE-PO employing RM as signal.

**LLM as Judge Reliablility**    To validate our choice of GPT-4o-mini as the LLM-as-a-judge, we conducted precisely this inter-rater reliability (IRR) analysis. We constructed an evaluation set by randomly sampling 200 generated examples. These examples were then assessed by three distinct sources: (1) human annotators, serving as the gold standard; (2) our selected judge, GPT-4o-mini; and (3) a stronger model, GPT-4o, for a comparative baseline. For the human annotation process, the 200 examples were distributed equally among four graduate student annotators, with each responsible for a unique batch of 50 examples The results are summarized in Table 4.

| Rater Pair Comparison | Agreement (%) | Cohen's Kappa ($\kappa$) |
|-----------------------|---------------|---------------------------|
| GPT-4o-mini vs. Human | 94.0% | 0.88 |
| GPT-4o vs. Human | 96.5% | 0.93 |
| GPT-4o-mini vs. GPT-4o | 98.5% | 0.97 |

Table 4: Inter-rater reliability results of different annotaters

# 4 CONCLUSION

We propose counterfactual-enhanced policy optimization that couples task accuracy with continuous causal signals, yielding smoother training and consistent gains across models and datasets, especially on Thinking backbones. The approach is a drop-in for PPO/GRPO and reduces reward-hacking behaviors, improving stability during training and robustness at convergence. Future work based on CE-PO could possible work on scale interventions and automate the accuracy–causality balance; CE-PO opens new optimization avenues by incorporating richer causal signals (e.g., Hessian-based beyond simple Jacobian) and enabling adaptive schemes that tune this trade-off on the fly.

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

## A  RELATED WORKS

### A.1  REWARD HACKING IN LLM

Reward hacking occurs when models exploit imperfections in reward functions rather than achieving the intended objective (Amodei et al., 2016; Everitt et al., 2017; Zhou et al., 2025). In large language models, this problem emerges prominently under alignment: proxy reward models can be gamed, leading to behaviors such as generating superficially persuasive but incorrect answers Wen et al. (2024), sycophancy Shrama et al. (2023), and exploiting evaluator biases in LLM-as-judge settings Liu et al. (2023); Wang et al. (2023). More recently, iterative refinement has revealed *in-context reward hacking*, where models adapt outputs to maximize evaluation scores rather than genuinely improving reasoning quality Pan et al. (2023; 2024). Mitigation strategies typically involve improved supervision (e.g., rationale-level feedback Ross et al. (2017)), evaluator debiasing, or structural interventions such as counterfactual augmentation Kaushik et al. (2020) and invariance penalties Arjovsky et al. (2020). However, these often rely on external annotations nor emphasized robustness–accuracy trade-offs. Our method introduces differentiable, counterfactual-enhanced causal signals as internal rewards, jointly optimizing accuracy and causal coherence to directly curb shortcut exploitation.

### A.2  CAUSAL INFERENCE AND REASONING IN LLMS

There have been several studies centering on causal reasoning capability in LLMs. Corr2Cause finds poor correlation-to-causation transfer by pure LLM reasoning (Jin et al., 2024), CoT often fails to be causally responsible for answers (Bao et al., 2024a), and faithful CoT remains challenging even with fine-tuning or activation edits (Tanneru et al., 2024). Methods that *impose* causal structure are emerging (e.g., $G^2$-Reasoner combines knowledge and goal prompts to improve causal reasoning) (Chi et al., 2025). Mechanistic approaches intervene directly in internal representations—editing factual circuits and testing hypotheses via causal/activation patching—to probe and modify pathways (Heimersheim et al., 2024). There also several studies utilizing SCM (Yu et al., 2025) and causal graph (Sheth et al., 2025) for better reasoning. Yet most studies are *post-hoc*. We build on this direction by using interventional, pathway-aware signals during post-training, and by explicitly aggregating them with task rewards to control the accuracy–causal-coherence trade-off.

## B  THEORETICAL GUARANTEES

**Setup.**  Let $\pi, \pi_{\text{old}}$ be stationary policies, $\gamma \in (0, 1)$ the discount factor, and $d^\pi$ the normalized discounted state distribution. Fix the baseline policy $\pi_{\text{old}}$ and let $A(s, a) = A_{\pi_{\text{old}}}(s, a)$ be its advantage under the CE-PO reward $r_{\text{CE}}$ (the Minkowski combiner of all channels). For each state $s$, denote by $\mathcal{N}_s \subset \mathbb{R}^{|\mathcal{A}|}$ the subspace spanned by *nuisance directions* (e.g., verbosity/length) estimated via counterfactual perturbations, and let $P_s$ be the Euclidean orthogonal projector onto $\mathcal{N}_s$. Write the action simplex $\Delta^{|\mathcal{A}|} = \{p \in \mathbb{R}^{|\mathcal{A}|} : p \geq 0, \ \mathbf{1}^\top p = 1\}$ with tangent space $\mathcal{T}_s = \{v \in \mathbb{R}^{|\mathcal{A}|} : \mathbf{1}^\top v = 0\}$. Define the *orthogonalized advantage*

$$A_\perp(s, \cdot) := (I - P_s)\, A(s, \cdot), \qquad \varepsilon_\perp := \max_s \|A_\perp(s, \cdot)\|_\infty.$$

and let

$$\alpha := \max_s D_{\text{TV}}\big(\pi(\cdot|s), \pi_{\text{old}}(\cdot|s)\big) = \tfrac{1}{2} \max_s \|\pi(\cdot|s) - \pi_{\text{old}}(\cdot|s)\|_1.$$

The TRPO-style surrogate is

$$L_{\pi_{\text{old}}}^{\text{CE}}(\pi) = \eta_{\text{CE}}(\pi_{\text{old}}) + \frac{1}{1 - \gamma} \mathbb{E}_{s \sim d^{\pi_{\text{old}}}} \big[ \ \langle A(s, \cdot), \ \pi(\cdot|s) - \pi_{\text{old}}(\cdot|s) \rangle \ \big],$$

noting that $\langle A(s, \cdot), \pi_{\text{old}}(\cdot|s) \rangle = 0$ for all $s$.

**Assumptions and Propositions**

    **P0 (Fixed reward at $\pi_{\text{old}}$)**  All CE components used to construct $r_{\text{CE}}$ (Jacobians, counterfactual subspaces $\mathcal{N}_s$, projectors $P_s$, and channel scores) are computed at $\pi_{\text{old}}$ and held fixed while evaluating $\eta_{\text{CE}}(\pi)$ over one update step.

**A0 (Residualization on the action-simplex tangent)** By construction of $r_{\mathrm{CE}}$ via counterfactual projection, the CE advantage is locally invariant along nuisance directions at $\pi_{\mathrm{old}}$:

$$\langle A(s, \cdot),\, v \rangle \;=\; 0 \quad \text{for all } v \in \mathcal{N}_s \cap \mathcal{T}_s.$$

Equivalently, $P_s A(s, \cdot) = 0$ and hence $A(s, \cdot) = A_\perp(s, \cdot) \in \mathcal{N}_s^\perp$.

**A1 (Well-posedness & bounded rewards)** Base rewards and channel scores are in $[0, 1]$ and CE weights sum to one, so $r_{\mathrm{CE}} \in [0, 1]$. The MDP is standard (finite or $\sigma$-finite), ensuring existence of $V^\pi, Q^\pi, A_\pi$.

**Theorem 1** (Orthogonalized TRPO Lower Bound for CE-PO). *For any $\pi$, with $\alpha = \max_s D_{\mathrm{TV}}\big(\pi(\cdot|s), \pi_{\mathrm{old}}(\cdot|s)\big)$, the CE-PO performance satisfies*

$$\eta_{\mathrm{CE}}(\pi) \;\geq\; L_{\pi_{\mathrm{old}}}^{\mathrm{CE}}(\pi) \;-\; \frac{4\gamma}{(1-\gamma)^2}\, \varepsilon_\perp\, \alpha^2. \tag{8}$$

*Proof.* **Step 1: Performance-difference decomposition.** The performance-difference lemma gives

$$\eta_{\mathrm{CE}}(\pi) - \eta_{\mathrm{CE}}(\pi_{\mathrm{old}}) = \frac{1}{1-\gamma}\, \mathbb{E}_{s \sim d^\pi}\big[\langle A(s, \cdot),\, \pi(\cdot|s)\rangle\big].$$

Add and subtract the surrogate:

$$\eta_{\mathrm{CE}}(\pi) = L_{\pi_{\mathrm{old}}}^{\mathrm{CE}}(\pi) + \frac{1}{1-\gamma}\, \mathbb{E}_s\Big[(d^\pi - d^{\pi_{\mathrm{old}}})(s)\, \langle A(s, \cdot),\, \pi(\cdot|s)\rangle\Big] =: L_{\pi_{\mathrm{old}}}^{\mathrm{CE}}(\pi) - \frac{1}{1-\gamma}T,$$

so it suffices to bound $|T|$.

**Step 2: Bounding the occupancy shift.** By a standard coupling argument, $\|d^\pi - d^{\pi_{\mathrm{old}}}\|_1 \leq \frac{2\gamma}{1-\gamma}\,\alpha$. Thus

$$|T| \leq \big(\max_s |\langle A(s, \cdot),\, \pi(\cdot|s)\rangle|\big)\, \|d^\pi - d^{\pi_{\mathrm{old}}}\|_1 \leq \frac{2\gamma}{1-\gamma}\,\alpha\, \max_s |\langle A(s, \cdot),\, \pi(\cdot|s)\rangle|.$$

**Step 3: Using CE-PO orthogonalization (A1).** For each $s$, by A1 we have $A(s, \cdot) = A_\perp(s, \cdot) \in \mathcal{N}_s^\perp$, and $\langle A(s, \cdot), \pi_{\mathrm{old}}(\cdot|s)\rangle = 0$ (definition of advantage). Hence

$$\langle A(s, \cdot),\, \pi(\cdot|s)\rangle = \langle A_\perp(s, \cdot),\, \pi(\cdot|s) - \pi_{\mathrm{old}}(\cdot|s)\rangle \leq \|A_\perp(s, \cdot)\|_\infty\, \|\pi(\cdot|s) - \pi_{\mathrm{old}}(\cdot|s)\|_1 \leq 2\,\varepsilon_\perp\,\alpha.$$

Taking the supremum over $s$ yields $\max_s |\langle A(s, \cdot),\, \pi(\cdot|s)\rangle| \leq 2\,\varepsilon_\perp\,\alpha$, and therefore

$$|T| \leq \frac{2\gamma}{1-\gamma}\,\alpha\,(2\,\varepsilon_\perp\,\alpha) = \frac{4\gamma}{1-\gamma}\,\varepsilon_\perp\,\alpha^2.$$

**Step 4: Final bound.** Plugging into Step 1,

$$\eta_{\mathrm{CE}}(\pi) \geq L_{\pi_{\mathrm{old}}}^{\mathrm{CE}}(\pi) - \frac{4\gamma}{(1-\gamma)^2}\,\varepsilon_\perp\,\alpha^2,$$

which is inequality (8). $\qquad\square$

**Properties.** *(Tightening via nuisance removal, $\ell_2$ statement).* Let $\varepsilon_{\mathrm{base}}^{(2)} := \max_s \|A_{\mathrm{base}}(s, \cdot)\|_2$ be the corresponding constant under a non-residualized baseline reward, and define $\varepsilon_\perp^{(2)} := \max_s \|A_\perp(s, \cdot)\|_2$. If there exists $\rho \in (0, 1]$ such that $\|P_s A_{\mathrm{base}}(s, \cdot)\|_2 \geq \rho\, \|A_{\mathrm{base}}(s, \cdot)\|_2$ for all $s$ *(nontrivial nuisance energy)*, then by Pythagoras for orthogonal projectors,

$$\varepsilon_\perp^{(2)} \;\leq\; \sqrt{1 - \rho^2}\, \varepsilon_{\mathrm{base}}^{(2)}.$$

Since $\|\cdot\|_\infty \leq \|\cdot\|_2$, we also have $\varepsilon_\perp \leq \varepsilon_\perp^{(2)}$, yielding a strictly smaller penalty constant in the $\ell_2$ analogue of equation 8 (and a looser but still improved constant in $\ell_\infty$ up to norm equivalence).

*(Null penalty for hacking-only updates).* If for some $s$ the per-state policy change lies entirely in $\mathcal{N}_s$, i.e., $\pi(\cdot|s) - \pi_{\mathrm{old}}(\cdot|s) \in \mathcal{N}_s$, then $(I - P_s)(\pi - \pi_{\mathrm{old}}) = 0$ and hence $\langle A_\perp(s, \cdot),\, \pi(\cdot|s) - \pi_{\mathrm{old}}(\cdot|s)\rangle = 0$, so that state contributes nothing to the penalty term in equation 8.

**KL-based corollaries.** By Pinsker's inequality $D_{\text{TV}}(p, q) \leq \sqrt{\frac{1}{2}\text{KL}(p\|q)}$, if $\delta := \max_s \text{KL}(\pi_{\text{old}}(\cdot|s)\|\pi(\cdot|s))$, then

$$\eta_{\text{CE}}(\pi) \geq L_{\pi_{\text{old}}}^{\text{CE}}(\pi) - \frac{2\gamma}{(1-\gamma)^2}\varepsilon_\perp \delta. \tag{9}$$

If only an *expected* per-state KL is enforced, $\mathbb{E}_{s\sim d^{\pi_{\text{old}}}}\big[\text{KL}(\pi_{\text{old}}\|\pi)\big] \leq \bar{\delta}$, then equation 9 holds with $\delta = C\bar{\delta}$ for a method-dependent constant $C \geq 1$ (e.g., via ratio clipping or per-state caps used in TRPO/PPO).

**From counterfactual Jacobians to action-space nuisance.** Let $\mathcal{U}_s$ be a finite set of representation-space directions obtained from counterfactual Jacobians (e.g., top singular vectors of counterfactual $J_{AB}$ blocks), and let $g_s$ be a smooth channel-score functional (e.g., spectral energy share). Write $G_s := \nabla_{\pi(\cdot|s)}\, g_s$ for its Jacobian with respect to action probabilities. Define

$$\mathcal{N}_s := \text{span}\big\{\, G_s^\top u \,:\, u \in \mathcal{U}_s \,\big\} \cap \mathcal{T}_s.$$

If each channel score is *residualized* by projecting $J_{AB}$ onto $\mathcal{U}_s^\perp$, then, at $\pi_{\text{old}}$, the resulting CE reward satisfies $\langle \nabla_\pi r_{\text{CE}}(s,\cdot),\, v \rangle = 0$ for all $v \in \mathcal{N}_s \cap \mathcal{T}_s$, implying $P_s A(s,\cdot) = 0$ (Assumption **A0**).

# C  ADDITIONAL TECHNIQUES

**Jacobian causal score is time and space-efficient in LLMs.** We design the Jacobian-based causal reward to be *practical at scale*: it reuses forward activations, avoids any explicit Jacobian/Hessian materialization, and replaces costly subspace operations with a constant-time projection. Concretely, we rely on three tricks. *(i) Local Jacobian via vJP (vector–Jacobian Projection).* Sensitivities are computed with one forward and two backward passes using vector–Jacobian products (reverse-mode autodiff), i.e., for a test direction $U$ we use $\text{vJP}(s_B, E(A); U) = \langle J_{AB}, U\rangle_F = J_{AB}^\top \text{vec}(U)$ and the scalar sensitivity $\|J_{AB}^\top \text{vec}(U)\|_2$; so no dense $J \in \mathbb{R}^{Td \times Td}$ (or Hessian) is ever formed; working memory stays at $\mathcal{O}(Td)$ for token embeddings (Baydin et al., 2018; Pearlmutter, 1994). *(ii) Residual projection.* The counterfactual direction is removed with a single inner product: if $g$ is the local Jacobian direction and $\hat{c}$ the (reshuffled) counterfactual direction, we use $r = g - (g^\top \hat{c})\hat{c}$, avoiding SVD/subspace construction and extra buffers (Golub and Van Loan, 2013). *(iii) Mixed precision.* Forward/backward in FP16/bfloat16 with dynamic loss scaling provides wall-clock gains while roughly halving activation memory, without changing the algorithmic interface (Micikevicius et al., 2018; NVIDIA Corporation, 2020).

*Throughput and footprint.* On a single modern GPU (e.g., A100 80 GB) and Llama-3-8B-Instruct, our implementation returns a reward in $\approx 2$ s, with negligible persistent memory beyond base activations. Simple Jacobian takes $O(Td)$ space complexity and would take $O(C_{fwd})$ time complexity, where $C_{fwd}$ means the time for one single forward in LLM reasoning. Yet for Hessian score, there needs to be continuous iteration of simple Jacobian to converge for a stable score and space complexity would also be $O((Td)^2)$, which is not practical. To provide a more quantitative analysis, we benchmarked our Jacobian computation against a BERTScore baseline on Qwen-3-14B (on a single node of 4 A100 GPU) using 100 randomly generated samples. We found the baseline took 1.20s and peaked at 28.00 GB, while our Jacobian step took 6.98s and peaked at 30.18 GB. This represents a 5.82x wall-clock time overhead and, more importantly, only a 1.078x GPU memory usage multiplier.

# D  ADDITIONAL PLOTS

**Question.** *Context: For husbands that don't set the alarm, the probability of ringing alarm is 90%. For husbands that set the alarm, the probability of ringing alarm is 25%. Question: For husbands that set the alarm, would it be less likely to see ringing alarm if the husband had not set the alarm?*

**Ground Truth:** No

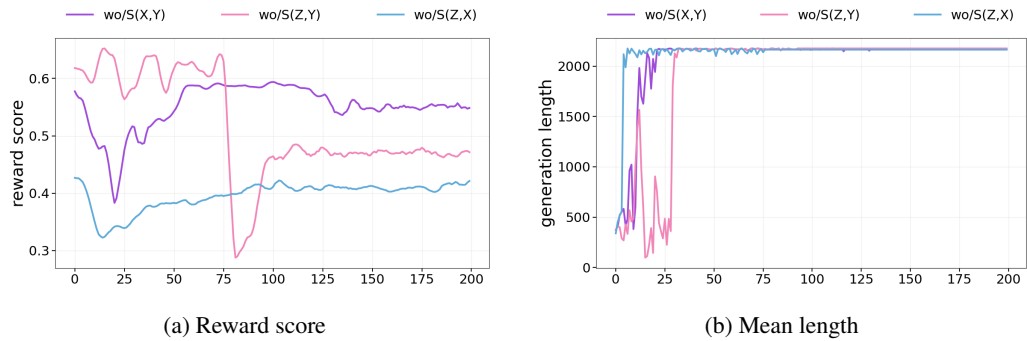

(a) Reward score          (b) Mean length

Figure 11: This figure supplements the ablation analysis and demonstrates that omitting a single Jacobian term induces reward degradation accompanied by length inflation.

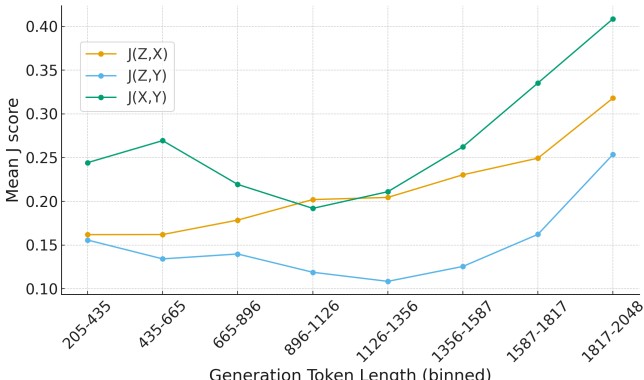

Figure 13: This figure illustrates the binned raw Jacobian scores distribution among different generation token length, echoing the reward hacking by length statement in Figure 2.

| Method | Prediction (Reasoning) |
|---|---|
| BERTScore | Predicts "Yes"; relies on surface overlap (*...compares 25% vs 90% directly but ignores that the subject of "less likely" is "not set"*). |
| Jacobian Only | Predicts "Yes"; observes 25% < 90% but *misplaces the subject, treating "set" as the focus of "less likely" instead of "not set"*. |
| CE-GRPO | Predicts "No"; correctly checks whether 90% < 25% (*false*) and concludes "not set" actually makes ringing more likely, matching ground truth. |

**Takeaways.**

- **Jacobian Only** tends to *template-follow*: latches onto "less likely ⇒ smaller %" and ignores subject/polarity flips, causing causal misread.
- **BERTScore** measures overlap, not causal/conditional structure; it cannot reason about which condition the "less likely" modifies.

Figure 12: Comparison of methods on the alarm-setting counterfactual example on Qwen-3-4B-Thinking.

## D.1 RESPONSE TEXT OF FIGURE 12

**CE-GRPO**: " <think> Okay, let's try to figure out this probability question. First, I need to understand the context. The problem says that for husbands who don't set the alarm, there's a 90% chance the alarm rings. For those who do set the alarm, the probability is 25%. The question is asking whether it's less likely to see the alarm ring if the husband had not set the alarm. Wait, that seems a bit confusing. Let me break it down.

The question is about the probability of the alarm ringing given whether the husband set the alarm or not. The context gives two probabilities: 90% for not setting the alarm and 25% for setting it. But the question is phrased a bit oddly. It says, "would it be less likely to see ringing alarm if the husband had not set the alarm?" Hmm. Maybe I need to rephrase that. Wait, maybe the question is asking if a husband didn't set the alarm, is it less likely that the alarm rings compared to if he did set it? But that's exactly what the context says. Because 90% is higher than 25%, so if the husband didn't set the alarm (90% chance), it's more likely the alarm rings. But the question is phrased as "less likely to see ringing alarm if the husband had not set the alarm," which would be the opposite. So maybe the answer is no, it's more likely. But maybe I'm misunderstanding the question. Let me check again. The context is about the probability of the alarm ringing given whether the husband set it. The question is asking if the husband didn't set the alarm, is it less likely to see the alarm ring? But according to the context, it's 90%, which is higher than 25%, so it's more likely. Therefore, the answer is no, it's not less likely; it's more likely. But maybe the question is trying to trick me into thinking about the reverse. Like, if the alarm doesn't ring, is it less likely the husband set it? But that's a different question. The original question is about the probability of the alarm ringing given whether the husband set it. So the answer is that it's more likely when the husband doesn't set the alarm. Therefore, the answer is no, it's not less likely. </think> The context states that husbands who **do not set the alarm** have a **90% probability** of the alarm ringing, while those who **do set the alarm** have a **25% probability**. The question asks whether it is **less likely** to see the alarm ring if the husband **did not set the alarm**. This is directly contradicted by the given probabilities: - **90% (high probability)** of the alarm ringing when the husband **does not set** the alarm. - **25% (low probability)** of the alarm ringing when the husband **does set** the alarm. Thus, it is **not less likely** to see the alarm ring if the husband did not set the alarm. Instead, it is **more likely** (90% vs. 25%). **Answer:** No, it is more likely to see the alarm ring if the husband did not set the alarm (90% vs. 25%)."

**BERTScore**: "<think> Okay, let's try to figure out this probability question. The user is asking whether it's less likely for a husband to ring the alarm if he had not set the alarm, compared to when he did set it. Wait, the question is a bit confusing. Let me parse it again. The context says: For husbands that don't set the alarm, the probability of ringing the alarm is 90%. For those that do set the alarm, it's 25%. The question is: For husbands that set the alarm, would it be less likely to see ringing alarm if the husband had not set the alarm? Hmm, that seems contradictory. Wait, maybe there's a typo or misunderstanding. Let me check again. Wait, the question is phrased as: "For husbands that set the alarm, would it be less likely to see ringing alarm if the husband had not set the alarm?" That doesn't make sense. If a husband has set the alarm, then he's in the "set the alarm" group. The question is asking if, for those who set the alarm, would it be less likely to ring the alarm if they hadn't set it? That seems like a paradox. Maybe the user meant to ask: For husbands that set the alarm, is it less likely to ring the alarm compared to those who didn't set it? Because the numbers are 25% vs 90%. So, if a husband sets the alarm, the probability of ringing is 25%, which is lower than 90% for those who don't set it. So, yes, it's less likely. But the way the question is phrased is confusing. Let me make sure I'm interpreting it correctly. Alternatively, maybe the question is trying to ask: If a husband didn't set the alarm, is it less likely to ring the alarm compared to if he had set it? But that's not what's written. The original question says: "For husbands that set the alarm, would it be less likely to see ringing alarm if the husband had not set the alarm?" So, it's a conditional. If the husband had not set the alarm, would it be less likely? But that's not a standard way to phrase it. Maybe the user is confused. Let me think again. Wait, maybe the question is trying to ask: For husbands who set the alarm, is it less likely that the alarm rings compared to those who don't set it? Because the numbers are 25% vs 90%. So, yes, setting the alarm reduces the probability of the alarm ringing. But the way the question is phrased is confusing. The user might be mixing up the conditions. Let me try to rephrase the question. The user might be asking: If a husband sets the alarm, is it less likely that the alarm rings compared to if he didn't set it? In that case, yes, because 25% is less than 90%. So the answer would be yes. But the original question is phrased as "For husbands that set the alarm, would it be less likely to see ringing alarm if the husband had not set the alarm?" Which is a bit of a paradox. Maybe it's a translation error or a misphrasing. But based on the numbers given, the answer is yes. So, the answer is yes, it's less likely. But I need to make sure I'm not misinterpreting. Let me check again. The question is: For husbands that set the alarm, would it be less likely to see ringing alarm if the husband had not set the alarm? So, if the husband had not set the alarm, would it be less likely? But that's not a standard question. Maybe the user meant: For husbands that set the alarm, is it less likely that the alarm rings compared to those who didn't set it? In that case, yes. So, the answer is yes. But the way the question is phrased is confusing. However, given the numbers, the answer is yes. </think> The question seems to have a slight ambiguity, but based on the context provided, here's the clarification: - **If a husband sets the alarm**, the probability of the alarm ringing is **25%**. - **If a husband does not set the alarm**, the probability of the alarm ringing is **90%**. Thus, **it is less likely** for the alarm to ring if the husband **set the alarm** (25%) compared to if he **did not set it** (90%). **Answer:** Yes, it is less likely for the alarm to ring if the husband set the alarm (25%) compared to if he did not (90%). The probability is lower when the alarm is set."

**Jacobian-Only**: "<think> Okay, let's try to figure out this probability question. So the context is about husbands and alarms. The problem says that for husbands who don't set the alarm, the probability of the alarm ringing is 90%. And for those who do set the alarm, the probability is 25%. The question is asking: For husbands that set the alarm, would it be less likely to see the alarm ring if the husband had not set the alarm? Wait, that seems a bit confusing. Let me parse that again. Wait, the question is phrased as: "For husbands that set the alarm, would it be less likely to see ringing alarm if the husband had not set the alarm?" Hmm. Maybe there's

a typo or confusion in the wording. Let me think. Maybe it's asking whether, given that a husband set the alarm, is it less likely that the alarm rings compared to if he hadn't set it? But that doesn't make sense because the probability of the alarm ringing is given based on whether the husband set it or not. So if a husband set the alarm, the probability is 25%, and if he didn't, it's 90%. So, if a husband set the alarm, the probability of the alarm ringing is 25%, which is lower than 90% if he hadn't set it. So, in that case, yes, it would be less likely to see the alarm ring if the husband had set it (25%) compared to if he hadn't (90%). But the question is phrased as "For husbands that set the alarm, would it be less likely to see ringing alarm if the husband had not set the alarm?" That seems contradictory. Wait, maybe it's a trick question. Let me try to rephrase. The question is asking: If a husband sets the alarm, is it less likely that the alarm rings compared to if he had not set the alarm? But that's not the case. Because if he set the alarm, the probability is 25%, and if he didn't, it's 90%. So, actually, it's more likely that the alarm rings if he didn't set it. So the answer would be no, it's not less likely. Wait, but the question is phrased as "would it be less likely to see ringing alarm if the husband had not set the alarm?" So maybe the question is trying to ask if, given that a husband set the alarm, is it less likely to see the alarm ring compared to if he had not set it. But that's not the case. The probability is lower when he sets the alarm. So the answer would be yes, it's less likely. But the wording is confusing. Let me check again. Original question: "For husbands that set the alarm, would it be less likely to see ringing alarm if the husband had not set the alarm?" So, the phrasing is a bit odd. Maybe it's trying to ask: If a husband set the alarm, is it less likely that the alarm rings compared to if he had not set the alarm? In that case, the answer is yes, because 25% is less than 90%. But the way the question is phrased is confusing. Maybe it's a trick question where they are trying to test understanding of conditional probability. Alternatively, maybe it's a case of confusing the two probabilities. Let me think again. The problem gives two probabilities: P(ring — not set) = 90%, and P(ring — set) = 25%. The question is asking: For husbands that set the alarm, would it be less likely to see ringing alarm if the husband had not set the alarm? Wait, that seems to be mixing up the conditions. If a husband set the alarm, then the probability of the alarm ringing is 25%. If he had not set the alarm, the probability is 90%. So, if we are considering a husband who set the alarm, the probability of the alarm ringing is 25%, which is less than 90% if he hadn't set it. Therefore, the answer is yes, it would be less likely. But the question is phrased in a way that seems to be asking if, given that the husband set the alarm, is it less likely to see the alarm ring compared to if he had not set it. So the answer is yes. But maybe the question is trying to trick you into thinking about it in a different way. Alternatively, maybe it's a case of confusing the two probabilities. Let me check if there's any other interpretation. Alternatively, maybe the question is asking: If a husband set the alarm, is it less likely that the alarm rings compared to if he had not set it? Which is exactly what the numbers say. So the answer is yes. But the way the question is phrased is confusing. Maybe the user made a typo. Alternatively, maybe it's a case of confusing the conditional probabilities. For example, if you have a husband who set the alarm, the probability of the alarm ringing is 25%, which is lower than 90% if he hadn't set it. Therefore, the answer is yes, it would be less likely. So the answer should be yes. But I need to make sure I'm not missing anything here. Let me think again. The question is phrased as: For husbands that set the alarm, would it be less likely to see ringing alarm if the husband had not set the alarm? So, the phrasing is a bit confusing. Maybe it's trying to ask: If a husband set the alarm, is it less likely that the alarm rings compared to if he had not set it? Which is the case. So the answer is yes. Therefore, the answer is yes, it would be less likely. But maybe the user is confused because they are mixing up the conditions. So the answer is yes, because the probability of the alarm ringing is 25% when the husband set it, which is less than 90% when he didn't. Therefore, for husbands that set the alarm, it is less likely to see the alarm ring compared to if they had not set it. So the answer is yes. </think> The question is asking whether, for husbands who set the alarm, it would be less likely to see the alarm ring if the husband had not set the alarm. **Key Information:** - **If a husband does not set the alarm**, the probability of the alarm ringing is **90%**. - **If a husband sets the alarm**, the probability of the alarm ringing is **25%**. **Analysis:** - The probability of the alarm ringing is **lower (25%)** when the husband sets the alarm compared to when he does not (90%). - Therefore, for husbands who set the alarm, it is **less likely** to see the alarm ring than if the husband had not set it. **Answer:** Yes, it would be less likely to see the alarm ring if the husband had not set the alarm. The probability of the alarm ringing is 25% when the husband sets it, which is significantly lower than the 90% probability when he does not set it."

# E  BROADER IMPACT

The proposed CE-PO framework has the potential to substantially improve the reliability and trustworthiness of large language models. By explicitly rewarding causal coherence in addition to task accuracy, the method reduces the tendency of models to exploit superficial cues or engage in reward hacking, which can lead to misleading or unfaithful reasoning. In practice, this enables LLMs to generate explanations and rationales that are not only correct in outcome but also grounded in meaningful reasoning processes—an essential property for high-stakes domains such as law, science, medicine, and education.

Beyond improving alignment, CE-PO can serve as a general methodology for building AI systems that are "right for the right reasons," supporting transparency, auditability, and robustness under distribution shifts. This contributes to safer deployment of generative models in real-world decision-making pipelines. At the same time, emphasizing causal regularization may help mitigate harmful biases that arise when models rely on spurious correlations, thereby supporting more equitable AI systems.

# F  PROMPT TEMPLATES

We presented the prompt template for clear $Z$, $X$, $Y$ separation in Figure 14 and detailed instruction for LLM-as-Judge in Figure 15.

---

**Prompt Instructions (Final Answer Tagging)**

```
"Solve the question and generate in assigned format."
"Wrap your final answer between <finalanswer> </finalanswer> tags.
"
"It's mandatory and required to not include anything after
</finalanswer> tag.  Don't let your generation process to redundant
and lengthy."
"Be very clear in your explanation, and ensure the final answer is
presented separately."
"Example of output:  We analyze A, solve B and answer is C. So final
answer is <finalanswer> C is correct </finalanswer>"
```

Figure 14: Verbatim instructions for final-answer tagging.

---

**Prompt Template (Strict Judge)**

**Task**
Decide if the final answer **matches** the ground truth.

**Rules (both must hold)**
1) **Final-answer equivalence**: extract the last explicit final answer and compare it to the ground truth *after trimming whitespace and ignoring case*.
2) **Reasoning validity** : Validity ($Z \rightarrow X \rightarrow Y$): Z must substantiate X, and Z,X must substantiate Y. Accept only reasoning that is logically sound, evidence-backed, and contradiction-free; apply near-adversarial scrutiny—any gaps, guesses, or inconsistencies $\rightarrow 0$.

**Question Z**
{question}

**Ground Truth Answer**
{correct_answer}

**Provided Reasoning X+Y**
{reasoning_process}

**Output (plain-text JSON, no code fences)**

```
{
  "judge_explanation": "<brief explanation>",
  "result": "<Yes or No>"
}
```

Figure 15: Prompt template for strict binary judgment requiring answer equivalence and valid reasoning.

---

Table 5: Welch's $t$-tests comparing $S(A, B)$ distributions between Incorrect and Correct groups.

| Metric | $t$ | $p$-value |
|--------|------|-----------|
| $S(Z, X)$ | 3.55 | 0.0005 |
| $S(X, Y)$ | 3.23 | 0.0014 |
| $S(Z, Y)$ | 3.15 | 0.0018 |

# G  DATASET FEATURES

### Table 6: Training Dataset

| Dataset | Question | Answer |
|---|---|---|
| **BBEHCausal** | Question: Alice (AF) and Bob (BF) each fire a bullet at a window, simultaneously striking the window. The window only shatters (WS) if it is hit by two bullets. Is Alice firing the bullet a sufficient cause for the window shattering? Reply based on the answer a logician would give. | No |
| **CasHOLD** | Context: limitations was equitably tolled fails because he did not raise this argument before the district court. See Hinton v. Pac. Enters., 5 F.3d 391, 395 (9th Cir.1993) (HOLDING). Grace's remaining contentions lack merit. Select the correct holding (1 to 4): "holding that the burden to allege facts sufficient to establish jurisdiction resides with plaintiff", "holding that the plaintiff bears the burden when relying on the discovery rule", "recognizing the validity of the doctrine but holding no equitable tolling on the facts presented", "holding that plaintiff bears the burden to timely allege facts supporting equitable tolling", "holding that the burden is on the plaintiff to allege facts sufficient to establish jurisdiction" | 3: "holding that plaintiff bears the burden to timely allege facts supporting equitable tolling" |
| **MATHHARD** | There are thirty-five red, yellow, orange, and white marbles in a bag. If half the number of red marbles equals two less than the number of yellow marbles, equals a third the number of orange marbles, and equals a third of three more than the number of white marbles, how many red marbles are there? | 8 |
| **IfQA** | If Caroline Flack's mother had felt the name suited her, what would have been Caroline's name? | Caroline |

# H  THE USE OF LARGE LANGUAGE MODELS (LLMS)

LLMs are the primary subject of our reinforcement learning (RL) experiments: we fine-tune pretrained models under CE-PO and baseline objectives to study stability, reward hacking, and causal alignment. Separately, we use LLMs as utilities for controlled text transformations—rephrasing, grammar correction, and during writing phase.

Table 7: Testing Dataset

| Dataset | Question | Answer |
|---|---|---|
| **BBEHMATH** | Consider the operations $a \; [] \; b \; = \; \begin{cases} (a-b), & a+b>0 \\ -a--b, & \text{otherwise} \end{cases}$, $a \; ; \; b \; = \; \begin{cases} (b-a)*b, & \|a-b\|<2 \\ (a-b)*a, & \text{otherwise} \end{cases}$, $a \; ! \; b \; = \; \begin{cases} (2\,[]\,b)*a, & a>b \\ b, & \text{otherwise} \end{cases}$, and $a \; @ \; b \; = \; \begin{cases} (a\,!\,b), & a-b>0 \\ a*b, & \text{otherwise} \end{cases}$; define $A = \big(((-5 \; ; \; -7) + (2 \; @ \; -4)) \; [] \; ((-9 \; [] \; -2) \; ! \; (6 \; \cdot \; 1))\big) \; [] \; \big(((7-6) \; [] \; (6 + -6)) \cdot ((-2 \; [] \; 9) \; ; \; (-1 - 5))\big)$, $B = \big(((9 \; ; \; 7) \; [] \; (6 \cdot 7)) \; [] \; ((2\,!\,-2) + (3 \; [] \; -6))\big) \; ; \; \big(((1-8)+(-5\,@\,-5)) \; ; \; ((-4-10) \; ; \; (-3\,@\,6))\big)$, and $C = \big(((-10\,!\,-1)\,!\,(-5+-6))\,@\,((-8+5)\cdot(-10+10))\big) \; ; \; \big(((-3-3) \; ; \; (2\,[]\,-5)) - ((2\cdot9)-(-8\,[]\,2))\big)$; compute $A + B - C$ (answer as a number). | -10038 |
| **CLadder** | Context: For husbands that don't set the alarm, the probability of ringing alarm is 42%. For husbands that set the alarm, the probability of ringing alarm is 51%. Question: For husbands that set the alarm, would it be less likely to see ringing alarm if the husband had not set the alarm? | yes |
| **LegalBench** | Context: Overall, the percentages and correlation co-efficients in nine of the ten endogenous are sufficient for purposes of the second Gingles precondition. See 478 U.S. at 56. Plaintiffs have shown minority political cohesiveness in these [**105] appellate-judgeship elections. Nonetheless, the stark statistic remains that, in Alabama's history, only two African Americans have been elected to statewide office for a total of [**186] three general elections and three primary elections. At the statewide level, this factor weighs in favor of Plaintiffs. Question: Based on the excerpt above, did the judge's legal reasoning rely on statistical evidence (e.g., regression analysis) when determining whether there was a causal link? Answer Yes or No | Yes |
| **LogiQA** | Some purple clay pots are alive. Therefore, some living things have good or bad quality.Question: Which of the following judgments, if true, would best strengthen the argument?Options: ['The quality of purple clay teapots is different from good to bad.', 'Some purple clay pots are inanimate.', 'There is no difference in the quality of purple clay teapots.', 'Some living things are not purple clay pots.'] | 0, The quality of purple clay teapots is different from good to bad. |

