# OpenReview forum: "Causally-Enhanced Reinforcement Policy Optimization of Large Language Models"
_ICLR.cc/2026/Conference — ICLR 2026 Conference Desk Rejected Submission_

### Official Review · Reviewer_x5eh · 2025-10-28

**Soundness:** 3
**Presentation:** 3
**Contribution:** 3
**Rating:** 8
**Confidence:** 2

**Summary:**

This paper aims to solve the prevalent problem of "reward hacking" in the reinforcement learning training of large LLMs. Specifically, models tend to generate answers that are superficially correct but are based on spurious or unfaithful reasoning. To address this, the paper proposes the Causally-Enhanced Policy Optimization (CE-PO) framework. The core of this method is to apply "reward-shaping" to the standard PPO/GRPO reward function by fusing scores from two dimensions: 1).  task accuracy e.g., using BERTScore.
2) Causal coherence, A novel, Jacobian-based proxy signal used to measure the internal influence along the generation pathway $Z \rightarrow X \rightarrow Y$ (prompt $\rightarrow$ rationale $\rightarrow$ answer).

The method's primary innovation is "Counterfactual Hardening". The authors argue that the raw Jacobian signal (gradient sensitivity) is easily "gamed" by superficial features (like length or format). Therefore, they construct counterfactual inputs (e.g., by permuting tokens) to estimate and project out these "nuisance cues," resulting in a more robust coherence score.

Finally, the paper uses a Minkowski power-mean combiner to flexibly fuse the accuracy and coherence scores, and this unified reward $R_p$ is used for PPO/GRPO training. The experiments claim that this method improves accuracy and robustness across multiple benchmarks.

**Strengths:**

1.  The problem the paper addresses—reward hacking and unfaithful reasoning in RL training—is one of the most central and pressing challenges in LLM alignment today. Its motivation holds very high research value.
2.  The paper does not stop at using raw gradient signals but acutely recognizes the fragility of the gradient signal itself (i.e., its susceptibility to being gamed by factors like length). The "Counterfactual Hardening" proposed to solve this is an intricate and novel idea. Attempting to separate semantic signals from nuisance signals via projection is an insightful approach.
3.  The experiment in Figure 2 strongly demonstrates (at least in the authors' setup) that an un-hardened, Jacobian-based reward (Non-CF-GRPO) quickly learns to "game" the system by increasing generation length.
4.  The ablation studies in Table 2 effectively demonstrate the necessity of the three Jacobian signals $S(Z,X)$, $S(X,Y)$, and $S(Z,Y)$ within the CE-PO framework. Removing any one of them leads to a catastrophic drop in performance, proving the importance of "closing the causal loop" within the authors' framework.

**Weaknesses:**

I am not an expert in the RL area. Therefore, the following comments are based on my current understanding, and I welcome any corrections to potential misunderstandings on my part.

1. The paper's title and text claim "Causally-Enhanced," but its core proxy is the Jacobian matrix $J_{AB}$. A Jacobian measures "local sensitivity," which is fundamentally a correlational metric, not a measure of causal intervention. A model might be sensitive to an input (high Jacobian score) merely because it learned a superficial shortcut, not because of true causal reliance. The paper lacks a comparison validating its proxy signal against causal measurement methods

2. The method's core assumption is that permuting tokens is sufficient to create a counterfactual that preserves "nuisance" cues while destroying "semantics." This definition is overly narrow and lacks justification. The paper provides no experiments showing why permutation is the best or only way, nor does it compare against other methods (e.g., deletion, replacement).

3. The method projects out nuisance signals, which implicitly relies on an unproven assumption: that the "semantic subspace" and "nuisance subspace" are (approximately) orthogonal. However, in deep networks, valid signals (like detailed reasoning) and superficial features (like long text) are likely highly entangled. Projecting out the "length" signal could very well "injure" the legitimate reasoning signal at the same time.

4. In RL training, reward calculation must be nearly instantaneous. To compute a single $R_p$, the method requires 3 Jacobians, and each of those requires $K=4$ counterfactual Jacobians to build the nuisance subspace. This means every reward calculation requires over a dozen gradient computations (VJPs), an extremely high overhead for practical training, especially on large models. The analysis of training cost will be helpful.

**Questions:**

No

---

> ### Author Response · Authors · 2025-11-17
> **Response to reviewer x5eh**
>
> We would like to thank you for such a constructive review and positive feedback, and we are glad to address the questions in response below.
>
> > Clarification of methodology:
>
> We thank the reviewer for the insightful methodological critique. The Jacobian matrix here is not perfectly plot causal effects. In fact, as illustrated in our paper from line 130 to 133, Jacobian scores are proxy to DCE (Direct Causal Effect). Intuitively, they can be understood as first-order Taylor approximations of the causal effect around the observed data point. With raw Jacobians, we do observe a superficial shortcut (e.g. by generation length), yet we have successfully projected out such nuisances factors via counterfactual hardening. Thus we can provide a stable and trustworthy causal effect proxy reward score.
>
> Our choice of permutation for counterfactual hardening was deliberate. We required a method that destroys semantics while perfectly preserving low level nuisance cues, in order to approximate nuisance dimensions of raw Jacobian. Alternative methods like deletion alter the 'length' cue. If replacing words with the same meaning words would fail to destroy the semantic sufficiently, and simply masking with [PAD] token would introduce new nuisances and alter token frequency. Under this target to approximate nuisance, permutation would most ideal sufficiently break semantics yet preserve the nuisance cues.
>
> Regarding the question of orthogonality assumption, actually this projection method was more supported empirically rather than assuming strict orthogonality.  We treat that "nuisance" and "semantic" subspaces as just sufficiently different for our projection to be beneficial.  If this "injury" were the dominant effect, our method's performance would decrease. Instead, our empirical results (e.g. Table 1) show consistent, significant performance gains, which empirically prove "decoupling" benefit far outweighs any potential "injury."
>
> >  Quantitative Analysis of Jacobian Score calculation:
>
> We are glad to provide additional experiments to further validate the efficiency of CE-PO design. Due to the limitation of computation resources, we can only include additional experiments on Qwen-3-14B at maximum and reviewer may also refer to Appendix C, which we have already provided on how Jacobian score is time and space efficient analysis (i,e. about 2 second for each Jacobian output on 4 A100 GPU)
>
> To provide a quantitative analysis, we benchmarked our Jacobian computation against a BERTScore baseline on Qwen-3-14B-Thinking (on a node of 4 A100 GPU) using 100 randomly selected samples of the test datasets. We found the baseline (BERTScore) took 1.20s and peaked at 28.00 GB per case, while our Jacobian step took 6.98s and peaked at 30.18 GB. This represents a 5.82x wall clock time overhead and, more importantly, only a 1.078x GPU memory usage multiplier.
>
> Additionally, we give theoretical proof of time and space complexity of Jacobian causal signals and clarification of why the Hessian score is way more time consuming and complex. As stated in our paper at line 900, simple Jacobian takes $O(Td)$ space complexity and would take $O(C_{fwd})$ time complexity, where $C_{fwd}$ means the time for one single forward in LLM reasoning. Yet for Hessian score, there needs to be continuous iteration of simple Jacobian to converge for a stable score and space complexity would also be $O((Td)^2)$, which is not practical.
>
>
> Thank you once again for your detailed feedback—it has been extremely helpful in improving our work. We hope our response strengthens your understanding and please don’t hesitate to reach out if you have any further questions. Thank you!

---

> ### Author Response · Authors · 2025-11-24
> **Official Comment by Authors**
>
> Dear reviewer, we thank you for your thoughtful and constructive review of our manuscript. Please let us know if you have any remaining concerns. We would be happy to provide any further clarification.

---

### Official Review · Reviewer_fWbQ · 2025-11-01

**Soundness:** 2
**Presentation:** 3
**Contribution:** 2
**Rating:** 4
**Confidence:** 3

**Summary:**

This paper proposes Causally-Enhanced Policy Optimization (CE-PO), a training-time reward shaping scheme for LLMs that augments standard PPO/GRPO with a differentiable proxy for causal coherence along the prompt-rationale-answer pathway. The method measures Jacobian-based sensitivities, hardens them with source-side counterfactual permutations to filter nuisance directions, and fuses the resulting scores with an accuracy reward via a power-mean combiner. Experiments report accuracy gains over PPO/GRPO baselines, reductions in length-driven reward hacking, and modest out-of-distribution improvements.

**Strengths:**

1. The paper presentation is clear and easy to read.

2. The paper turns internal, differentiable signals (block Jacobians of span log-likelihoods with respect to upstream token embeddings + counterfactual hardening) into a scalar reward, which is a clear path making “how” matter during RL.

3. The proposed reward shaping can be plugged into PPO/GRPO without architectural changes, and the power mean combiner exposes a single knob to trade accuracy against coherence.

**Weaknesses:**

1. **Dependence on BERTScore as the base accuracy reward:** BERTScore is convenient and differentiable, but it is not a task-ground truth for many reasoning datasets and can mis-rank semantically brittle outputs. Results with verifiable rewards in common reasoning-heavy domains like math and code should be included.

2. **Beyond length hacking:** The paper shows resistance to length-based reward hacking, which aligns with known reward-model over-optimization effects. However, claims in the paper, e.g., in abstract "reduces unfaithful chain-of-thought" is not well-supported by results beyond length hacking.

3. **Faithfulness of gradient proxies:** The method relies on first-order Jacobians as a stand-in for directed causal effects. These signals are known to be fragile in some contexts, and the projection-based “direct-effect” isolation could over- or under-correct depending on geometry and scale [1].

[1] Basu, Samyadeep, Phil Pope, and Soheil Feizi. "Influence Functions in Deep Learning Are Fragile." International Conference on Learning Representations.

**Questions:**

See Weaknesses.

---

> ### Author Response · Authors · 2025-11-17
> **Response to reviewer fWbQ (Part I)**
>
> We appreciate your detailed feedback and reflective comments. To further clarify and tackle your concerns, we provide responses below by turn.
>
> > Usage of BERTScore:
>
> We thank the reviewer for this critical point regarding our baseline, and we justify the usage of BERTScore below and provide additional experiments to further scale up our design of CE-PO. First, there exists technical necessity: our Minkowski combiner (Eq. 7) typically **prefers** a continuous, [0, 1] $r_{base}$ to stably fuse with our dense causal signals. As noted (Lines 375 to 376), standard 0/1 rewards are "less compatible" and "induce oscillatory updates." as detailed with LLM as Judge reward in Ablation Studies.
>
> Second, BERTScore is also a commonly used semantic metric in RL for LLM, like in RL4LM [1] specifies BERTScore as valid semantic reward function, and BERTScore is also used as reward function in paper [2] and [3].However, to fully address the reviewer's valid concern about baseline strength and to demonstrate the generality of our framework, we have added two new sets of experiments in the revised version.
>
> We replace BERTScore with a standard pretrained Reward Model (RM) as $r_\text{base}$, over here we consider using DeBERTa-v3-large to make it applicable on our computing resource. This provides a more conventional baseline, yet generally they all provide a continuous scalar to measure how much the generated answer approximates the ground truth ones in embedding space.
>
> We added results on two new, highly reasoning intensive datasets (CodeXGLUE [4] and HARP [5]), which are respectively code detect dataset and human annotated math reasoning problems and included the result table below.
>
> | Model | Method | BBEHMATH | CLadder | LegalBench | LogiQA | Average |
> | :--- | :--- | :---: | :---: | :---: | :---: | :---: |
> | Qwen-3-4B-Thinking | GRPO-RM | 6.61 | 44.44 | 77.33 | 81.03 | 52.85 |
> |  | CE(RM)-GRPO | 8.39 | 52.37 | 77.15 | 83.70 | 55.40 |
> | Phi-3.5-mini-Ins. | GRPO-RM | 7.18 | 40.85 | 69.29 | 55.52 | 43.71 |
> |  | CE(RM)-GRPO | 7.29 | 48.91 | 70.33 | 58.16 | 46.17 |
> | Llama-3-8B-Ins. | GRPO-RM | 8.39 | 39.54 | 63.00 | 69.58 | 45.13 |
> |  | CE(RM)-GRPO | 14.79 | 45.00 | 67.35 | 72.62 | 49.94 |
>
> | Dataset/Llama-3-8B-Ins. | Vanilla | CE-GRPO | CE-PPO | GRPO-BERTScore | PPO-BERTScore |
> | :--- | :---: | :---: | :---: | :---: | :---: |
> | CodeXGLUE | 30.67 | 40.00 | 38.00 | 34.67 | 33.33 |
> | HARP | 28.67 | 31.33 | 32.67 | 30.00 | 31.33 |
>
> [1] Ramamurthy, Rajkumar, et al. "Is reinforcement learning (not) for natural language processing: Benchmarks, baselines, and building blocks for natural language policy optimization." arXiv preprint arXiv:2210.01241 (2022).
>
> [2] Zhao, Shuai, et al. "Learning from reference answers: Versatile language model alignment without binary human preference data." arXiv preprint arXiv:2504.09895 (2025).
>
> [3] Huang, Jiawei, et al. "Can rlhf be more efficient with imperfect reward models? a policy coverage perspective." arXiv preprint arXiv:2502.19255 (2025).
>
> [4] Lu, Shuai, et al. "Codexglue: A machine learning benchmark dataset for code understanding and generation." arXiv preprint arXiv:2102.04664 (2021).
>
> [5] Yue, Albert S., et al. "HARP: A challenging human-annotated math reasoning benchmark." arXiv preprint arXiv:2412.08819 (2024).

---

> > ### Author Response · Authors · 2025-11-17
> > **Response to reviewer fWbQ (Part II)**
> >
> > > Clarification of our Contribution (length hacking just necessitates counterfactual enhancement)
> >
> > We thank the reviewer for this comment, and we would like to clarify our contribution, which is far more than tackle length reward hacking in using non-counterfactual Jacobian scores. Figure 2 was intended only to demonstrate how one component of our method (Counterfactual Hardening) solves the superficial "length hacking" symptom caused by the raw, non-CF reward . Our central claim about "reducing unfaithful chain of thought" is supported by more direct evidence elsewhere: namely, Figure 8, which shows our method is the only one to simultaneously improve accuracy and all three causal coherence signals (the very definition of faithfulness) , and Figure 12, which provides a concrete qualitative example where our model follows the correct causal logic while the baseline takes an unfaithful shortcut. Additionally, main experiments at Table 1,2 and Figure 4,5  etc further validate that CE-PO has excelled performance, satisfactory performance and OOD robustness.
> >
> > > Faithfulness of gradient proxies:
> >
> > We thank the reviewer for this insightful critique. We agree that Jacobians are only a first order approximation of directed causal effects, yet it has been proven as faithful and useful in approximating causal effects in neural networks [1][2].  Additionally, another paper[3] shows that raw Jacobian in signal fidelity is very close to the ground truth effect.
> >
> > We acknowledge this limitation (Lines 152–159) and, since first-order Jacobians already perform well in our setting, we see no practical need here to utilize higher-order (e.g., Hessian-based) estimators and leave it to future works. Importantly, we provide targeted experiments showing that Jacobian based scores are both necessary and effective in our setting. Our ablations in Table 2 and Figure 11 are particularly revealing: the reviewer is right to worry that the projection based “direct effect” isolation (S(Z,Y)) might over- or under-correct, but when we remove any one of the Jacobian based signals (including (S(Z,Y)); see Table 2, row 5), the model’s average performance collapses catastrophically. This demonstrates that, although approximate, these signals are far from “fragile” in practice and are empirically crucial for mitigating reward hacking.
> >
> > Moreover, in ablation studies of this paper where we inject Gaussian noise with ($\sigma$ = 0.05) into the reward, the overall performance degrades only mildly (within a few points), further indicating that our gradient based proxy is both faithful to the underlying causal structure and robust to moderate perturbations of the reward signal.
> >
> >  Thank you once again for your detailed feedback—it has been extremely helpful in improving our work. Please don’t hesitate to reach out if you have any further questions. Thank you!
> >
> > [1] Chattopadhyay, Aditya, et al. "Neural network attributions: a causal perspective." International Conference on Machine Learning. PMLR, 2019.
> >
> > [2] Kancheti, Sai Srinivas, et al. "Matching learned causal effects of neural networks with domain priors." arXiv preprint arXiv:2111.12490 (2021).
> >
> > [3] Yeh, Chih-Kuan, et al. "On the (in) fidelity and sensitivity of explanations." Advances in neural information processing systems 32 (2019).

---

> ### Author Response · Authors · 2025-11-24
> **Official Comment by Authors**
>
> Dear reviewer, we thank you for your thoughtful and constructive review of our manuscript. Please let us know if you have any remaining concerns. We would be happy to provide any further clarification.

---

> ### Author Response · Authors · 2025-11-27
> **Gentle Reminder**
>
> Dear Reviewer,
>
> Thank you very much for your time and thoughtful feedback. We fully understand that you may have a busy schedule. With less than one week remaining in the discussion period, we would be deeply grateful if you could take a moment to review our responses and let us know whether they sufficiently address your concerns.
>
> We truly appreciate your time and consideration.

---

### Official Review · Reviewer_5jTM · 2025-11-01

**Soundness:** 3
**Presentation:** 3
**Contribution:** 3
**Rating:** 6
**Confidence:** 3

**Summary:**

The paper introduces Causally-Enhanced Policy Optimization (CE-PO), a novel, drop-in reward-shaping framework designed to mitigate reward hacking and unfaithful reasoning in LLMs trained with RL. The method augments the standard policy objective (like PPO/GRPO) with a differentiable proxy for causal coherence along the prompt-to-rationale-to-answer ($Z \rightarrow X \rightarrow Y$) generation pathway. This is achieved by computing Jacobian-based internal influence signals, applying counterfactual hardening via residualization to remove spurious sensitivities (e.g., to output length) , and finally fusing the coherence score with task accuracy using a tunable Minkowski (power-mean) combiner. Experiments show that CE-PO successfully reduces reward hacking, improves robustness to causal perturbations, and yields an average accuracy improvement of 5.49% over baselines on reasoning benchmarks.

**Strengths:**

The paper proposes a novel, differentiable, and annotation-free method using Jacobian-based sensitivities and counterfactual hardening to create an internal proxy for causal coherence across the $Z \rightarrow X \rightarrow Y$ generation path.

The core technique of counterfactual hardening successfully and empirically prevents reward hacking from spurious non-semantic factors like generation length, a known major issue in outcome-centric RL for LLMs.

CE-PO demonstrates consistent performance gains, improving average accuracy by 5.49% across four datasets and multiple LLM backbones, while also improving robustness to causal stress tests.

**Weaknesses:**

Calculating the block Jacobians and performing the counterfactual hardening process, especially with multiple source-side counterfactuals ($K$), significantly increases the computational cost of the reward function computation compared to a simple task accuracy reward.

The composite reward structure, which relies on a Minkowski combiner with four weighted terms and a tunable exponent $p$, introduces high complexity and potential sensitivity to the extensive hyperparameter search required for the optimal weights ($w_i$) and $p$.

The core coherence signal relies on Jacobians, which are only a first-order Taylor approximation of the Direct Causal Effect (DCE). While computationally efficient, this approximation may limit the precision of the causal alignment and potentially restrict the model's ability to capture complex, higher-order causal dependencies.

**Questions:**

Can the authors provide a quantitative analysis (e.g., wall-clock time overhead or GPU memory usage multiplier) of the Jacobian and counterfactual hardening steps when scaling up to much larger models (e.g., >30B parameters) where gradient computation becomes significantly more resource-intensive?

The Minkowski combiner introduces four weights ($w_i$) and the exponent $p$. Is there a principled method or guideline for selecting these hyperparameters to achieve a desired trade-off, or must this be treated as a purely empirical tuning problem for each new task and model architecture?

---

> ### Author Response · Authors · 2025-11-17
> **Response to reviewer 5jTM**
>
> We sincerely thank you for the helpful feedback, detailed responses and recognition of our work. To further clarify on the concerns, we provide responses below.
>
> > Response to weaknesses:
>
> We thank the reviewer for highlighting these key challenges in using internal rewards such as Jacobian scores; we address each point in turn below.
>
> 1. We designed the reward computation to be highly efficient: as detailed in Appendix C, we avoid materializing the full Jacobian by using efficient vector Jacobian products (vJP), keeping the overhead to only 2 seconds per case on an A100 node.
>
> 2. Similarly, the Minkowski combiner's complexity is an intentional feature for balancing the accuracy coherence trade-off. Yet, our main results (Table 1) show its robustness, as a single default configuration template works consistently well across all 5 models. In addition, ablation studies analyze when combiner parameters are set differently, how the trade-off and the optimization target shall be impacted.
>
> 3. It is correct that the Jacobian is a first order approximation, a limitation we explicitly discuss (Lines 258 to 260)  as a feasible solution against the high cost and instability of higher order estimators. The ablation studies (Table 2 and Figure 11) provide the crucial empirical validation: when any of the causal Jacobian signal is removed, performance catastrophically collapses , proving it is an essential and highly effective proxy.
>
> > Quantitative Analysis of Jacobian Score calculation cost:
>
> We are glad to provide additional experiments to further validate the efficiency of CE-PO design. Due to the limitation of computation resources, we can only include additional experiments on Qwen-3-14B at maximum and reviewer may also refer to Appendix C, which we have already provided on how Jacobian score is time and space efficient analysis (i,e. about 2 second for each Jacobian output on 4 A100 GPU)
>
> To provide a quantitative analysis, we benchmarked our Jacobian computation against a BERTScore baseline on Qwen-3-14B-Thinking (on a node of 4 A100 GPU) using 100 randomly selected samples of the test datasets. We found the baseline (BERTScore) took 1.20s and peaked at 28.00 GB per case, while our Jacobian step took 6.98s and peaked at 30.18 GB. This represents a 5.82x wall clock time overhead and, more importantly, only a 1.078x GPU memory usage multiplier.
>
> Additionally, we give theoretical proof of time and space complexity of Jacobian causal signals and clarification of why the Hessian score is way more time consuming and complex. As stated in our paper at line 900, simple Jacobian takes $O(Td)$ space complexity and would take $O(C_{fwd})$ time complexity, where $C_{fwd}$ means the time for one single forward in LLM reasoning. Yet for Hessian score, there needs to be continuous iteration of simple Jacobian to converge for a stable score and space complexity would also be $O((Td)^2)$, which is not practical.
>
>
> > Principle of hyperparameter selection of Minkowski combiner:
>
> Our hyperparameter choices are grounded in principled design rather than arbitrary tuning. As described in the paper, the exponent ppp functions as an interpretable control parameter: higher $p$ values place stronger emphasis on accuracy, whereas lower $p$ deliberately promotes causal coherence. Similarly, the 70% accuracy / 30% coherence weighting reflects a principled regularization strategy. The ablation results in Table 2 demonstrate that removing this 30% coherence component leads to substantial degradation, underscoring its necessity.
> Equally important, the overall framework exhibits low sensitivity to hyperparameter variation. The strongest evidence is that a single, fixed default configuration is applied across all main experiments in Table 1, yet it yields consistent improvements across five models and four datasets. Same supporting evidence can be found at ablation studies Table 2, where perturbed reward and setting weight as [1/2 , 1/6, 1/6, 1/6] all result in satisfactory performance.
>
>  Thank you once again for your detailed feedback—it has been extremely helpful in improving our work. Please don’t hesitate to reach out if you have any further questions. Thank you!

---

> > ### Author Response · Authors · 2025-11-24
> > **Official Comment by Authors**
> >
> > Dear reviewer, we thank you for your thoughtful and constructive review of our manuscript. Please let us know if you have any remaining concerns. We would be happy to provide any further clarification.

---

> > > ### Comment · Reviewer_5jTM · 2025-11-27
> > >
> > > Thanks for the reply and I'll maintain my positive score.

---

> > > > ### Author Response · Authors · 2025-11-27
> > > > **Official Comment by Authors**
> > > >
> > > > Thank you for your thoughtful feedback and the positive assessment. We sincerely appreciate your time and will take your comments and suggestions into account for polishing our submission.

---

### Official Review · Reviewer_oskw · 2025-11-03

**Soundness:** 2
**Presentation:** 4
**Contribution:** 3
**Rating:** 4
**Confidence:** 3

**Summary:**

The paper introduces Causally Enhanced Policy Optimization, which is a reward framework that modifies policy optimization with a proxy for causal coherence. The authors explain how they use counterfactual hardens and a power-mean combiner to improve this modification for empirical use, and test on a series of models to find improvements in both multistep arithmetic and more general domains such as legal and QA problems.

**Strengths:**

The paper is well written and the motivation for coherence is reasonable. Furthermore, the question tackled makes important steps towards improving a core aspect of LLMs - CoT faithfulness.

The modifications (e.g., counterfactual hardening) demonstrates that the authors have put significant efforts into investigating this space in order to make it work.

**Weaknesses:**

1. The current chain is Z -> X -> Y. However, rationale may sometimes exist independently of question; e.g., rules such as common sense. Can the authors clarify how this interacts with the training signal?

2. For the results in Figure 2, can you also show that high-length outputs simply have higher rewards, instead of just showing that the chosen generations are longer? This would directly confirm the findings that there is reward hacking.

3. Line 285, "To avoid random guesses": If I understand correctly, this is also because you need to generate X

4. For your experiments, you are using GPT-4o-mini for LLM as a judge: can you measure inter rater reliability with either a stronger model or some human judgments?

5. Your core baseline is PPO and GRPO with BERTScore reward. To the extent of my knowledge, this isn't a common reward signal used for these optimization procedures. This is illustrated by PPO and GRPO baselines reducing accuracy on truthfulQA (Figure 5). Can you justify your choice of BERTScore, or are there other baselines you can consider?

**Questions:**

See weaknesses.

---

> ### Author Response · Authors · 2025-11-17
> **Response to reviewer oskw (Part I)**
>
> We sincerely thank you for the thoughtful and constructive feedback, and detailed responses. We hereby provide clarification to the concerns and questions as below.
>
> > Further illustration of Z,X,Y coherence motivation:
>
> We thank the reviewer for this insightful question. CE-PO is designed as a prescriptive model, which means by calculating S(Z,X) we can distinguish between a "bad" agnostic template and a "good" faithfully applied universal logic (both independent rationales). Signal S(Z,X) rewards X's sensitivity to Z's specific details. For example, a "bad" X might be only a generic template (e.g., "Let me think...") followed by a shortcut answer $Y$. This Z-agnostic $X$ is penalized, whereas a "good" general reasoning logic applied to $Z$'s details is rewarded even if the reasoning rationale is common sense as long as it’s dependent on Z. This is proven with example in Figure 12: the baseline used a "bad" generic shortcut (comparing percentages) and failed, while CE-PO, forced by S(Z, X) to be sensitive to Z's specific causal structure, succeeded
>
> > Reward hacking of raw Jacobians:
>
> We thank the reviewer for the helpful comment, to further illustrate how raw Jacobian is length dependent. Overhere, we provide both theoretical and empirical analysis of the correlation between raw Jacobian and generation length.
>
> **Theoretical deduction for raw Jacobian  (no spectral normalization/counterfactual):**
>
> - Case 1: When $A$'s length ($T_A$) increases (Dimensional Hacking)
> By definition, $J_{AB} = \frac{\partial s_B}{\partial E(A)}$. Since $E(A)$ is a $T_A \times d$ matrix, $J_{AB}$ must also be a $T_A \times d$ matrix. The not normalized raw reward (norm) is:
> $R_{raw} =  \sqrt{\sum_{i=1}^{T_A} \sum_{j=1}^{d} (J_{ij})^2}$
> As $T_A$ increases, the number of terms in the outer summation $\sum_{i=1}^{T_A}$ mechanically increases, thus inflating the total value of $R_{raw}$.
>
> - Case 2: When $B$'s length ($T_B$) increases (Additive Hacking)
> The score $s_B$ is a sum over $B$'s tokens: $s_B = \sum_{t=1}^{T_B} \log p(b_t)$. By the sum rule of differentiation, $J_{AB}$ is also a sum:
> $$J_{AB} = \frac{\partial \left( \sum_{t=1}^{T_B} \log p(b_t) \right)}{\partial E(A)} = \sum_{t=1}^{T_B} \left( \frac{\partial \log p(b_t)}{\partial E(A)} \right)$$
> As $T_B$ increases, the final Jacobian $J_{AB}$ becomes the sum of more individual Jacobian matrices, which mechanically increases its overall magnitude (norm).
>
> **Additional experiments of Jacobian on different length generations:**
>
> We have included an additional figure illustrating the binned mean Jacobian score as a function of generation length, which was calculated using the reward logs from our CE-PO experiments on Llama-3-8B, which is included in our revised version of the paper detailed in Appendix D (Figure 13)
>
> > LLM as Judge reliability:
>
> We are glad to provide additional experiments of how LLM-as-Judge results are reliable under our setting via  inter rater reliability (IRR) analysis, which is also added in the revision version of the paper(in line 497 to 509). Yet generally, LLM-as-Judge is a reliable criteria proven in many literatures [1].
> We constructed an evaluation set by randomly sampling 200 generated examples. These examples were then assessed by three distinct sources: (1) human annotators; (2) our selected judge, GPT-4o-mini; and (3) a stronger model, GPT-4o, for a comparative baseline. For the human annotation process, the 200 examples were distributed equally among four graduate student annotators, with each responsible for a unique batch of 50 examples
> The results are summarized in Table below.
> | Rater Pair Comparison | Agreement (%) | Cohen's Kappa ($\kappa$) |
> | :--- | :---: | :---: |
> | GPT-4o-mini vs. Human | 94.0% | 0.88 |
> | GPT-4o vs. Human | 96.5% | 0.93 |
> | GPT-4o-mini vs. GPT-4o | 98.5% | 0.97 |
>
> [1] Zheng, Lianmin, et al. "Judging llm-as-a-judge with mt-bench and chatbot arena." Advances in neural information processing systems 36 (2023): 46595-46623.

---

> ### Author Response · Authors · 2025-11-17
> **Response to reviewer oskw (Part II)**
>
> > Usage of BERTScore:
>
> We thank the reviewer for this critical point regarding our baseline, and we justify the usage of BERTScore below and provide additional experiments to further scale up our design of CE-PO. First, there exists technical necessity: our Minkowski combiner (Eq. 7) typically **prefers** a continuous, [0, 1] $r_\text{base}$ to stably fuse with our dense causal signals. As noted (Lines 226 to 232 note), standard 0/1 rewards are "less compatible" and "induce oscillatory updates." as detailed with LLM as Judge reward in Ablation Studies.
>
> Second, BERTScore is also a commonly used semantic metric in RL for LLM, like in RL4LM [1] specifies BERTScore as valid semantic reward function, and BERTScore is also used as reward function in paper [2] and [3]. However, to fully address the reviewer's valid concern about baseline strength and to demonstrate the generality of our framework, we have added two new sets of experiments in the revised version.
>
>
> We replace BERTScore with a standard pretrained Reward Model (RM) as $r_{base}$, over here we consider using DeBERTa-v3-large to make it applicable on our computing resource. This provides a more conventional baseline, yet generally they all provide a continuous scalar to measure how much the generated answer approximates the ground truth ones in embedding space.
> We added results on two new, highly reasoning intensive datasets (CodeXGLUE [4] and HARP [5]), which are respectively code detect dataset and human annotated math reasoning problems.
> | Model | Method | BBEHMATH | CLadder | LegalBench | LogiQA | Average |
> | :--- | :--- | :---: | :---: | :---: | :---: | :---: |
> | Qwen-3-4B-Thinking | GRPO-RM | 6.61 | 44.44 | 77.33 | 81.03 | 52.85 |
> |  | CE(RM)-GRPO | 8.39 | 52.37 | 77.15 | 83.70 | 55.40 |
> | Phi-3.5-mini-Ins. | GRPO-RM | 7.18 | 40.85 | 69.29 | 55.52 | 43.71 |
> |  | CE(RM)-GRPO | 7.29 | 48.91 | 70.33 | 58.16 | 46.17 |
> | Llama-3-8B-Ins. | GRPO-RM | 8.39 | 39.54 | 63.00 | 69.58 | 45.13 |
> |  | CE(RM)-GRPO | 14.79 | 45.00 | 67.35 | 72.62 | 49.94 |
>
> | Dataset/Llama-3-8B-Ins. | Vanilla | CE-GRPO | CE-PPO | GRPO-BERTScore | PPO-BERTScore |
> | :--- | :---: | :---: | :---: | :---: | :---: |
> | CodeXGLUE | 30.67 | 40.00 | 38.00 | 34.67 | 33.33 |
> | HARP | 28.67 | 31.33 | 32.67 | 30.00 | 31.33 |
>
> Eventually, it’s indeed that statement in line 285 is employed to generate “X”, thank you for pointing that out and we have revised it in the paper.
>
> Thank you once again for your detailed feedback—it has been extremely helpful in improving our work. Please don’t hesitate to reach out if you have any further questions. Thank you!
>
> [1] Ramamurthy, Rajkumar, et al. "Is reinforcement learning (not) for natural language processing: Benchmarks, baselines, and building blocks for natural language policy optimization." arXiv preprint arXiv:2210.01241 (2022).
>
> [2] Zhao, Shuai, et al. "Learning from reference answers: Versatile language model alignment without binary human preference data." arXiv preprint arXiv:2504.09895 (2025).
>
> [3] Huang, Jiawei, et al. "Can rlhf be more efficient with imperfect reward models? a policy coverage perspective." arXiv preprint arXiv:2502.19255 (2025).
>
> [4] Lu, Shuai, et al. "Codexglue: A machine learning benchmark dataset for code understanding and generation." arXiv preprint arXiv:2102.04664 (2021).
> [5] Yue, Albert S., et al. "HARP: A challenging human-annotated math reasoning benchmark." arXiv preprint arXiv:2412.08819 (2024).

---

> > ### Comment · Reviewer_oskw · 2025-11-20
> >
> > I thank the authors for their efforts in this rebuttal. Especially the IRR is very helpful for contextualizing the quality of signals from LLM as a judge.
> >
> > My main remaining concern is about the last point on BERTScore. The first point about technical necessity, while justifying using a metric that is continuous and bounded, does not constrain options to only BERTScore. Further, while prior work has used BERTScore, this does not particularly explain decreases in performance on TruthfulQA, and more generally why paraphrase-type similarity is desired as the choice for PPO/GRPO in this setting.
> >
> > For the replacement reward model, DeBERTa-v3-large, how was it trained? My worry is that this gives the same kind of signal as BERTScore (embedding-based similarity to reference). Could you also provide whether this version still sees reductions on TruthfulQA?
> >
> > Currently, this concern is only partially resolved.
> >
> > Small point, it would be good to include original reasoning traces in figure 12.

---

> > > ### Author Response · Authors · 2025-11-24
> > > **Official Comment by Authors**
> > >
> > > Dear reviewer, we thank you for your thoughtful and constructive review of our manuscript. Please let us know if you have any remaining concerns. We would be happy to provide any further clarification.

---

> > > > ### Author Response · Authors · 2025-11-27
> > > > **Gentle Reminder**
> > > >
> > > > Dear Reviewer,
> > > >
> > > > Thank you very much for your time and thoughtful feedback. We fully understand that you may have a busy schedule. With less than one week remaining in the discussion period, we would be deeply grateful if you could take a moment to review our responses and let us know whether they sufficiently address your concerns.
> > > >
> > > > We truly appreciate your time and consideration.

---

> ### Author Response · Authors · 2025-11-21
> **Official Comment by Authors**
>
> Thank you for your response and we are glad to further clarify on your concerns.
>
> > Further clarification of BERTScore usage
>
> First, we clarify that BERTScore is not the only viable option for our reward signal. We selected it because, in our experiments, it provided stable fusion with our causal signals, whereas high SNR (Signal Noise Ratio) judge-based rewards led to unstable causal metrics fusion results. Our general preference is simply for a continuous, differentiable, and relatively low-SNR measure of closeness to the ground truth that can be stably fused with causal metrics. We do not use BERTScore exclusively, as pre-trained reward model (as shown in previous response) also provides satisfactory result. In the future, we would like to test to other metrics like ROUGE or Edit Distance. Yet it's not practical to traverse all possible metrics, thus our contribution lies in providing feasible yet not exclusive factual correctness metric option(s).
>
> Second, the mild decrease observed on TruthfulQA with PPO/GRPO+BERTScore is expected because TruthfulQA is an OOD benchmark that evaluates factual robustness or trustworthiness, which does not align perfectly with semantic accuracy. Additionally, decrease of BERTScore is not consistent, as GRPO-BERTScore result in not significant performance gap compared to vanilla model, which are respectively 60.71% and 61.56% out of 817 test cases. When we replace BERTScore with a DeBERTa reward model that provides a finer-grained factual-accuracy signal, this mild decrease does not appear as in table below, indicating that the issue arises from BERTScore being a bit more coarse similarity metric compared to pre-trained reward model. Yet it's not a limitation of our approach, and CE-PO integrates effectively with both BERTScore and DeBERTa-v3-large and excels the baselines and vanilla.
>
> | TruthfulQA   | Vanilla | GRPO-RM | CE(RM)-GRPO |
> | ------------ | ------- | ------- | ----------- |
> | Accuracy (%) | 61.56   | 62.54   | 63.64       |
>
> > More details of pre-trained reward model
>
> We would like to clarify more of the details of the supplementary experiment. Over here, we use the pretrained DeBERTa-v3-large model [1], trained on large-scale internet corpora, as an encoder and map its pooled representation to a scalar correctness score for each (prompt, answer) pair, rather than computing cosine similarity to a reference as done in BERTScore.
>
> [1] https://huggingface.co/microsoft/deberta-v3-large
>
> Additionally, we have updated the whole response content of observation in Figure 12 at Appendix D.1 in revised version. Thank you for your suggestion!
>
> Please let us know if this addresses your concerns, and feel free to point out any further questions.

---

### Author Response · Authors · 2025-11-17
**General response**

Dear reviewers and AC:

We would like to thank the reviewers for their insightful feedback, which has provided us with excellent guidance towards crafting a more comprehensive and competitive paper! In the revised version, all changes are highlighted in blue. As highlighted by the reviewers, CE-PO directly addresses reward hacking and spurious correlation of CoT reasoning in RL for LLMs by explicitly promoting causal coherence along the prompt–rationale–answer pathway. We are encouraged that they find our framework “clearly written and well-motivated” , appreciate the novel Jacobian-based and counterfactually hardened plug in reward shaping, and stable consistent accuracy and robustness gains across models and reasoning benchmarks.

We are committed to sincerely addressing all of the reviewers’ concerns and have provided general responses to each of your comments below. In light of the feedback, we believe the enhancements we’ve made will resonate even more profoundly with our audience and meet an even higher standard, positioning our work to inspire a broader readership.

(1) Quantitative analysis of the computation cost of Jacobian scores (reviewer 5jTM & x5eh)

We compute Jacobian-based scores via efficient vector–Jacobian products (vJPs), rather than materializing full Jacobian tensors. As detailed in Appendix C, this yields $O(Td)$ space and $O(C_{fwd})$ time complexity, while Hessian-based scores would require repeated Jacobian evaluations and $O((Td)^2)$ space, which is impractical.
Empirically, on Qwen-3-14B-Thinking (4×A100, 100 samples), a BERTScore baseline takes 1.20s / 28.00 GB per case, whereas our Jacobian step takes 6.98s / 29.77 GB (5.82× wall-clock overhead but only a 1.063× memory multiplier). These results indicate that CE-PO’s causal Jacobian signals remain practical for large-scale training.


(2) Usage of BERTScore as factual accuracy may be less common in previous literatures

We clarify that CE-PO’s Minkowski combiner (Eq. 7) prefers a continuous base reward $\(r_{\text{base}}\in[0,1]\)$: our ablations studies (Table 2) show that discrete 0/1 LLM-as-judge rewards lead to unstable and ineffective learning. We also add references showing that BERTScore has already been successfully used as a reward model in RL for language models (e.g. RL4LM).

To further demonstrate the scalability of CE-PO pipeline, we added new experiments where we replace BERTScore by a standard pretrained reward model (DeBERTa-v3-large) as $r_{\text{base}}$. Across three models (Qwen-3-4B-Thinking, Phi-3.5-mini-Instruct, Llama-3-8B-Instruct) and four benchmarks as in Table 1. CE(RM)-GRPO consistently outperforms GRPO-RM (see the new table in the rebuttal), demonstrating that CE-PO’s gains are not tied to BERTScore.

Additionally, we also add results on two more challenging, reasoning-centric datasets (CodeXGLUE and HARP). CE-GRPO and CE-PPO consistently outperform both vanilla training and BERTScore-based GRPO/PPO, showing the scalability of more various datasets.

(3) Clarification of motivation, methodology, and contributions

Reviewers raised concerns about our motivation (e.g. design of Z-X-Y diagram in cases like common sense rationale by oskw), methodology (Jacobian faithfulness and necessity by fWbQ) , and overall contribution (results are beyond length hacking and reshuffling counterfactual validity by x5eh and fWbQ). We now clarify point by point.

Motivation:  we now clarify that CE-PO is not merely “scoring CoT” but *prescribing* faithful reasoning by explicitly separating input \(Z\), rationale \(X\), and answer \(Y\) and jointly optimizing \(S(Z,X)\), \(S(Z,Y)\), and \(S(X,Y)\) to enforce input-dependent, causally grounded, and self-consistent chains of thought.

Methodology: Jacobian scores are used as first-order Taylor approximations to direct causal effects, chosen as a practical alternative to computation expensive and unstable higher-order estimators. Empirically, they are both necessary and robust: ablating any Jacobian term leads to collapse, whereas adding Gaussian noise or moderate reweighting only mildly harms performance.

Contributions: Our counterfactual hardening is designed to preserve nuisance statistics that drive reward hacking while destroying semantics; in this setting, permutation is more appropriate than deletion, semantic replacement, or [PAD] masking, and our projection step is justified by its observed gains. Conceptually, length hacking is only one visible symptom; our main contribution is a causal coherence objective that aligns training with faithful CoT. CE-PO consistently improves both accuracy and all three coherence signals across models and benchmarks, and continues to outperform strong GRPO/PPO baselines even when replacing BERTScore with a reward model and moving to harder datasets (Figure 8, Table 1–2, Figure 5).


Thank you very much.

with best Regards,

Authors

---

### Note · Program_Chairs · 2026-01-17
**Submission Desk Rejected by Program Chairs**

The following references in this submission do not refer to real documents and/or have major errors in bibliographic information:

 Fan Bao et al. Non-causal reasoners in large language models. arXiv preprint arXiv:2407.01489, 2024a.
Boxin Wang et al. Large language models are not fair evaluators. arXiv preprint arXiv:2310.05492, 2023.
Sören Heimersheim et al. How to use and interpret activation patching. arXiv preprint arXiv:2405.05964, 2024.